# Into the Wild with AudioScope: Unsupervised Audio-Visual Separation of On-Screen Sounds

**Efthymios Tzinis**[*1], **Scott Wisdom**[2], **Aren Jansen**[2], **Shawn Hershey**[2], **Tal Remez**[2],
**Daniel P.W. Ellis**[2], **John R. Hershey**[2]
[1]University of Illinois at Urbana-Champaign    [2]Google Research
etzinis2@illinois.edu    scottwisdom@google.com

## Abstract

Recent progress in deep learning has enabled many advances in sound separation and visual scene understanding. However, extracting sound sources which are apparent in natural videos remains an open problem. In this work, we present *AudioScope*, a novel audio-visual sound separation framework that can be trained without supervision to isolate on-screen sound sources from real in-the-wild videos. Prior audio-visual separation work assumed artificial limitations on the domain of sound classes (e.g., to speech or music), constrained the number of sources, and required strong sound separation or visual segmentation labels. AudioScope overcomes these limitations, operating on an open domain of sounds, with variable numbers of sources, and without labels or prior visual segmentation. The training procedure for AudioScope uses mixture invariant training (MixIT) to separate synthetic mixtures of mixtures (MoMs) into individual sources, where noisy labels for mixtures are provided by an unsupervised audio-visual coincidence model. Using the noisy labels, along with attention between video and audio features, AudioScope learns to identify audio-visual similarity and to suppress off-screen sounds. We demonstrate the effectiveness of our approach using a dataset of video clips extracted from open-domain YFCC100m video data. This dataset contains a wide diversity of sound classes recorded in unconstrained conditions, making the application of previous methods unsuitable. For evaluation and semi-supervised experiments, we collected human labels for presence of on-screen and off-screen sounds on a small subset of clips.

## 1 Introduction

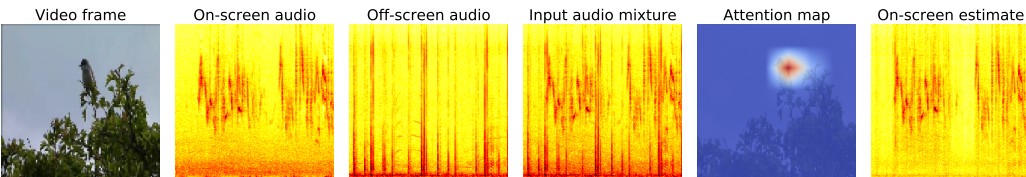

Figure 1: AudioScope separating on-screen bird chirping from wind noise and off-screen sounds from fireworks and human laugh. More demos online at `https://audioscope.github.io`.

Audio-visual machine perception has been undergoing a renaissance in recent years driven by advances in large-scale deep learning. A motivating observation is the interplay in human perception between auditory and visual perception. We understand the world by parsing it into the objects that are the sources of the audio and visual signals we can perceive. However, the sounds and sights produced by these sources have rather different and complementary properties. Objects may make sounds intermittently, whereas their visual appearance is typically persistent. The visual percepts of different objects tend to be spatially distinct, whereas sounds from different sources can blend together and overlap in a single signal, making it difficult to separately perceive the individual sources.

---

*Work done during an internship at Google.

This suggests that there is something to be gained by aligning our audio and visual percepts: if we can identify which audio signals correspond to which visual objects, we can selectively attend to an object's audio signal by visually selecting the object.

This intuition motivates using vision as an interface for audio processing, where a primary problem is to selectively preserve desired sounds, while removing unwanted sounds. In some tasks, such as speech enhancement, the desired sounds can be selected by their class: speech versus non-speech in this case. In an open-domain setting, the selection of desired sounds is at the user's discretion. This presents a user-interface problem: it is challenging to select sources in an efficient way using audio. This problem can be greatly simplified in the audio-visual case if we use video selection as a proxy for audio selection, for example, by selecting sounds from on-screen objects, and removing off-screen sounds. Recent work has used video for selection and separation of speech (Ephrat et al., 2018; Afouras et al., 2020) or music (Zhao et al., 2018; Gao & Grauman, 2019; Gan et al., 2020). However, systems that address this for arbitrary sounds (Gao et al., 2018; Rouditchenko et al., 2019; Owens & Efros, 2018) may be useful in more general cases, such as video recording, where the sounds of interest cannot be defined in advance.

The problem of associating arbitrary sounds with their visual objects is challenging in an open domain. Several complications arise that have not been fully addressed by previous work. First, a large amount of training data is needed in order to cover the space of possible sound. Supervised methods require labeled examples where isolated on-screen sounds are known. The resulting data collection and labeling burden limits the amount and quality of available data. To overcome this, we propose an unsupervised approach using mixture invariant training (MixIT) (Wisdom et al., 2020), that can learn to separate individual sources from in-the-wild videos, where the on-screen and off-screen sounds are unknown. Another problem is that different audio sources may correspond to a dynamic set of on-screen objects in arbitrary spatial locations. We accommodate this by using attention mechanisms that align each hypothesized audio source with the different spatial and temporal positions of the corresponding objects in the video. Finally we need to determine which audio sources appear on screen, in the absence of strong labels. This is handled using a weakly trained classifier for sources based on audio and video embeddings produced by the attention mechanism.

## 2 RELATION TO PREVIOUS WORK

Separation of arbitrary sounds from a mixture, known as "universal sound separation," was recently shown to be possible with a fixed number of sounds (Kavalerov et al., 2019). Conditional information about which sound classes are present can improve separation performance (Tzinis et al., 2020). The FUSS dataset (Wisdom et al., 2021) expanded the scope to separate a variable number of sounds, in order to handle more realistic data. A framework has also been proposed where specific sound classes can be extracted from input sound mixtures (Ochiai et al., 2020). These approaches require curated data containing isolated sounds for training, which prevents their application to truly open-domain data and introduces difficulties such as annotation cost, accurate simulation of realistic acoustic mixtures, and biased datasets.

To avoid these issues, a number of recent works have proposed replacing the strong supervision of reference source signals with weak supervision labels from related modalities such as sound class (Pishdadian et al., 2020; Kong et al., 2020), visual input (Gao & Grauman, 2019), or spatial location from multi-microphone recordings (Tzinis et al., 2019; Seetharaman et al., 2019; Drude et al., 2019). Most recently, Wisdom et al. (2020) proposed mixture invariant training (MixIT), which provides a purely unsupervised source separation framework for a variable number of latent sources.

A variety of research has laid the groundwork towards solving audio-visual on-screen source separation (Michelsanti et al., 2020). Generally, the two main approaches are to use audio-visual localization (Hershey & Movellan, 2000; Senocak et al., 2018; Wu et al., 2019; Afouras et al., 2020), or object detection networks, either supervised (Ephrat et al., 2018; Gao & Grauman, 2019; Gan et al., 2020) or unsupervised (Zhao et al., 2018), to predict visual conditioning information. However, these works only consider restricted domains such as speech (Hershey & Casey, 2002; Ephrat et al., 2018; Afouras et al., 2020) or music (Zhao et al., 2018; Gao & Grauman, 2019; Gan et al., 2020). Gao et al. (2018) reported results with videos from a wide domain, but relied on supervised visual object detectors, which precludes learning about the appearance of sound sources outside of a closed set of classes defined by the detectors. Rouditchenko et al. (2019) proposed a system for a wide domain of sounds,

but required sound class labels as well as isolated sounds from these classes. Our approach avoids the supervision of class labels and isolated sources in order to handle unknown visual and sound classes occurring in multi-source data.

Towards learning directly from a less restrictive open domain of in-the-wild video data, Tian et al. (2018) learned to localize audio-visual events in unconstrained videos and presented an ad hoc dataset. Korbar et al. (2018) pretrained models to discern temporal synchronization of audio-video pairs, and demonstrated promising results on action recognition and audio classification. Arandjelovic & Zisserman (2017) took a similar approach by classifying audio-visual correspondences of pairs of one video frame and one second of audio. Hu et al. (2020) proposed a curriculum learning approach where the model gradually learns harder examples to separate.

Closest to our work is the approach of Owens & Efros (2018), a self-supervised audio-visual on-screen speech separation system based on temporal audio-visual alignment. However, Owens & Efros (2018) assumes training videos containing only on-screen sources, and it is unclear how to adapt it to the case where training videos include off-screen sources.

Our approach significantly differs from these prior works in that we do not restrict our domain to musical instruments or human speakers, and we train and test with real in-the-wild videos containing an arbitrary number of objects with no object class restrictions. Our proposed framework can deal with noisy labels (e.g. videos with no on-screen sounds), operate on a completely open-domain of in-the-wild videos, and effectively isolate sounds coming from on-screen objects.

We address the following task, which extends the formulation of the on-screen speech separation problem (Owens & Efros, 2018). Given an input video, the goal is to separate all sources that constitute the input mixture, and then estimate an audio-visual correspondence score for each separated source. These probability scores should be high for separated sources which are apparent on-screen, and low otherwise. The separated audio sources, weighted by their estimated on-screen probabilities, can be summed together to reconstruct the on-screen mixture. We emphasize that our approach is more generally applicable than previous proposals, because real-world videos may contain an unknown number of both on-screen and off-screen sources belonging to an undefined ontology of classes.

We make the following contributions in this paper:

1. We provide the first solution for training an unsupervised, open-domain, audio-visual on-screen separation system from scratch on real in-the-wild video data, with no requirement on modules such as object detectors that require supervised data.

2. We develop a new dataset for the on-screen audio-visual separation task, drawn from 2,500 hours of unlabeled videos from YFCC100m, and 55 hours of videos that are human-labeled for presence of on-screen and off-screen sounds.

## 3 MODEL ARCHITECTURE

The overall architecture of AudioScope is built from the following blocks: an image embedding network, an audio separation network, an audio embedding network, an audio-visual attention mechanism, and an on-screen classifier (see Figure 2). The separation and embedding networks are based on prior work and are described in the following subsections. However, the main focus of this work is the overall architecture, as well as the training framework and loss functions.

The video is analyzed with the image embedding network, which generates local embeddings for each of 64 locations within each frame, as well as an embedding of the whole frame. These embeddings are used both as a conditioning input to an audio separation network, as well as an input for classification of the on-screen sounds. The audio separation network takes the mixed input waveform as input, and generates a fixed number of output waveforms, a variable number of which are non-zero depending on the estimated number of sources in the mixture. Conditioning on the video enables the separation to take advantage of cues about the sources present when performing separation. The audio embedding network is applied to each estimated source to obtain one embedding per frame for each source. These audio embeddings are then pooled over time and used in the audio-visual spatio-temporal attention network to retrieve, for each source, a representation of the visual activity that best matches the

audio, similar to the associative maps extracted from the internal network representations proposed by Harwath et al. (2018).

The architecture is designed to address the problem of unsupervised learning on in-the-wild open-domain data. First, because the target training videos can contain both on-screen and off-screen sounds, training a system to directly produce the audio of the target video would encourage inclusion of off-screen sounds as well as on-screen ones[1]. Our proposed multi-source separation network instead produces latent source estimates using an unsupervised MixIT objective, which has been shown to perform well at general sound separation (Wisdom et al., 2020). By decoupling separation from on-screen classification, our architecture facilitates the use of robust objectives that allow some of the sources to be considered off-screen, even if they appear in the soundtrack of the target videos.

The audio-visual attention architecture is motivated by the alignment problem between audio and video: sound source objects in video may be localized, may move over time, and may be present before and after the corresponding audio activity. Because of the open domain we cannot rely on a pre-defined set of object detectors to anchor the video representations of on-screen sources, as is done in some prior works (Ephrat et al., 2018; Gao & Grauman, 2019; Gan et al., 2020). Instead we propose attention to find the video representations that correspond to a source in a more flexible way.

The proposed strategy of temporal pooling of the audio embeddings, before using them in the spatio-temporal attention, allows the network to derive embeddings that represent the active segments of the source audio, and ignore the ambiguous silent regions. In the present model, video is analyzed at a low frame rate, and so the audio-visual correspondence is likely based on relatively static properties of the objects, rather than the synchrony of their motion with the audio. In this case, a single time-invariant representation of the audio may be sufficient as a proof of concept. However, in future work, with higher video frame rates, it may be worthwhile to consider using attention to align sequences of audio and video embeddings in order to detect synchrony in their activity patterns.

The on-screen classifier operates on an audio embedding for one estimated source, as well as the video embedding retrieved by the spatio-temporal attention mechanism, using a dense network. This presumably allows detection of the congruence between the embeddings. To provide additional context for this decision, a global video embedding, produced by temporal pooling, is provided as an additional input. Many alternative choices are possible for this classifier design, which we leave for future work, such as using a more complex classification architecture, or providing additional audio embeddings as input.

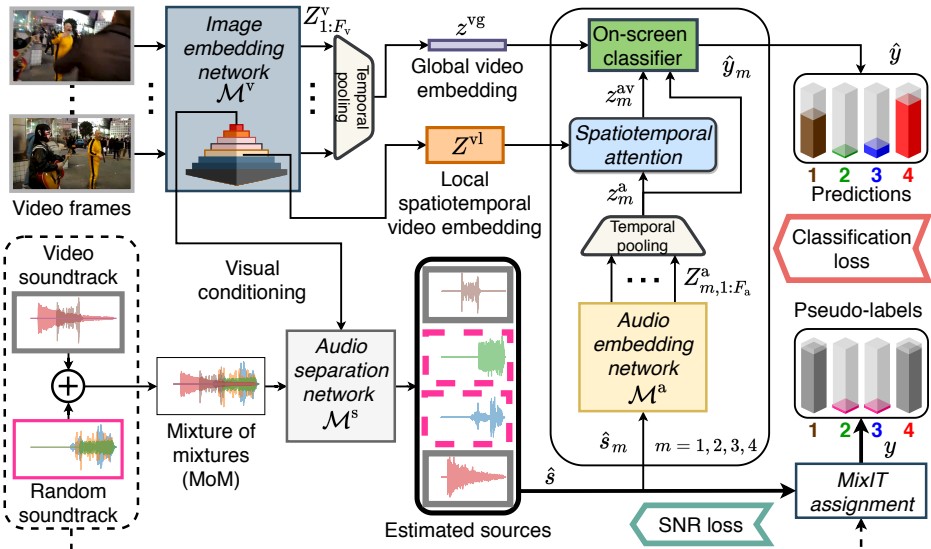

Figure 2: AudioScope system diagram with 2 input mixtures and 4 output sources.

---

[1]We train such a system in Appendix A.3.5, and find that it is not an effective approach.

## 3.1 AUDIO SEPARATION NETWORK

The separation network $\mathcal{M}^s$ architecture consists of learnable convolutional encoder and decoder layers with an improved time-domain convolutional network (TDCN++) masking network (Wisdom et al., 2020). A mixture consistency projection (Wisdom et al., 2019) is applied to constrain separated sources to add up to the input mixture. The separation network processes a $T$-sample input mixture waveform and outputs $M$ estimated sources $\hat{s} \in \mathbb{R}^{M \times T}$. Internally, the network estimates $M$ masks which are multiplied with the activations of the encoded input mixture. The time-domain signals $\hat{s}$ are computed by applying the decoder, a transposed convolutional layer, to the masked coefficients.

## 3.2 AUDIO EMBEDDING NETWORK

For each separated source $\hat{s}_m$, we extract a corresponding global audio embedding using the MobileNet v1 architecture (Howard et al., 2017) which consists of stacked 2D separable dilated convolutional blocks with a dense layer at the end. This network $\mathcal{M}^a$ first computes log Mel-scale spectrograms with $F_a$ audio frames from the time-domain separated sources, and then applies stacks of depthwise separable convolutions to produce the $F_a \times N$ embedding matrix $Z_m^a$, which contains an $N$-dimensional row embedding for each frame. An attentional pooling operation (Girdhar & Ramanan, 2017) is used, for each source, $m$, to form a static audio embedding vector $z_m^a = \mathrm{attend}(\bar{Z}_m^a, Z_m^a, Z_m^a)$, where the average embedding $\bar{Z}_m^a = \frac{1}{F_a} \sum_i Z_{m,i}^a$ is the query vector for source $m$. The attention mechanism (Bahdanau et al., 2015) is defined as follows:

$$\mathrm{attend}(q, K, V) = \alpha^T f_V(V), \quad \alpha = \mathrm{softmax}(\tanh(f_K(K)) \tanh(f_q(q))^T), \tag{1}$$

with query row vector $q$, the attention weight distribution column vector $\alpha$, key matrix $K$, value matrix $V$, and trainable row-wise dense layers $f_q$, $f_V$, $f_K$, all having conforming dimensions.

## 3.3 IMAGE EMBEDDING NETWORK

To extract visual features from video frames, we again use a MobileNet v1 architecture. This visual embedding model $\mathcal{M}^v$ is applied independently to each one of the $F_v$ input video frames and a static-length embedding is extracted for each image $Z_j^v$, $j \in \{1, \ldots, F_v\}$.

**Conditioning separation network with the temporal video embedding**: The embeddings of the video input $Z_j^v$ can be used to condition the separation network (Tzinis et al., 2020). Specifically, the image embeddings are fed through a dense layer, and a simple nearest neighbor upsampling matches the time dimension to the time dimension of the intermediate separation network activations. These upsampled and transformed image embeddings are concatenated with the intermediate TDCN++ activations and fed as input to the separation network layers.

**Global video embedding**: A global embedding of the video input is extracted using attentional pooling over all video frames, given by $z^{vg} = \mathrm{attend}(\bar{Z}^v, Z^v, Z^v)$, where the average embedding $\bar{Z}^v = \frac{1}{F_v} \sum_j Z_j^v$ is the query vector.

**Local spatio-temporal video embedding**: We also use local features extracted from an intermediate level in the visual convolutional network, that has $8 \times 8$ spatial locations. These are denoted $Z_k^{vl}$, where $k = (j, n)$ indexes video frame $j$ and spatial location index $n$. These provide spatial features for identification of sources with visual objects to be used with audio-visual spatio-temporal attention.

## 3.4 AUDIO-VISUAL SPATIO-TEMPORAL ATTENTION

An important aspect of this work is to combine audio and visual information in order to infer correspondence between each separated source and the relevant objects in video. This in turn will be used to identify which sources are visible on-screen. To this end, we employ an audio-visual spatio-temporal attention scheme by letting the network attend to the local features of the visual embeddings for each separated source. In this mechanism, we use the audio embedding $z_m^a$ as the query input for source $m$, and the key and value inputs are given by the spatio-temporal video embeddings, $Z^{vl}$. As a result, the flattened version of the output spatio-temporal embedding, corresponding to the $m$-th source, is $z_m^{av} = \mathrm{attend}(z_m^a, Z^{vl}, Z^{vl})$.

### 3.5 ON-SCREEN CLASSIFIER

To infer the visual presence each separated source, we concatenate the global video embedding $z^{\mathrm{vg}}$, the global audio embedding for each source $z_m^{\mathrm{a}}$, and the corresponding local spatio-temporal audio-visual embedding $z_m^{\mathrm{av}}$. The concatenated vector is fed through a dense layer $f_{\mathrm{C}}$ with a logistic activation: $\hat{y}_m = \mathrm{logistic}\left(f_{\mathrm{C}}\left([z^{\mathrm{vg}}, z_m^{\mathrm{a}}, z_m^{\mathrm{av}}]\right)\right)$.

### 3.6 SEPARATION LOSS

We use a MixIT separation loss (Wisdom et al., 2020), which optimizes the assignment of $M$ estimated sources $\hat{s} = \mathcal{M}^{\mathrm{s}}\left(x_1 + x_2\right)$ to two reference mixtures $x_1$, $x_2$ as follows:

$$\mathcal{L}_{\mathrm{sep}}\left(x_1, x_2, \hat{s}\right) = \min_A \left(\mathcal{L}_{\mathrm{SNR}}\left(x_1, [A\hat{s}]_1\right) + \mathcal{L}_{\mathrm{SNR}}\left(x_2, [A\hat{s}]_2\right)\right), \tag{2}$$

where the mixing matrix $A \in \mathbb{B}^{2 \times M}$ is constrained to the set of $2 \times M$ binary matrices where each column sums to 1. Due to the constraints on $A$, each source $\hat{s}_m$ can only be assigned to one reference mixture. The SNR loss for an estimated signal $\hat{t} \in \mathbb{R}^T$ and a target signal $t \in \mathbb{R}^T$ is defined as:

$$\mathcal{L}_{\mathrm{SNR}}(t, \hat{t}) = 10 \log_{10}\left(\|t - \hat{t}\|^2 + 10^{-3}\|t\|^2\right). \tag{3}$$

### 3.7 CLASSIFICATION LOSS

To train the on-screen classifier, we consider the following classification losses. These losses use the binary labels $y_m$, where are given for supervised examples, and in the unsupervised case $y_m = A_{1,m}^*$ for each source $m$, where $A^*$ is the optimial mixing matrix found by the minimization in (2). We also use the notation $\mathcal{R} = \{m | y_m = 1, \ m \in \{1, \ldots, M\}\}$ to denote the set of positive labels.

**Exact binary cross entropy**:

$$\mathcal{L}_{\mathrm{exact}}\left(y, \hat{y}\right) = \sum_{m=1}^{M}\left(-y_m \log\left(\hat{y}_m\right) + (y_m - 1)\log\left(1 - \hat{y}_m\right)\right). \tag{4}$$

**Multiple-instance cross entropy**: Since some separated sources assigned to the on-screen mixture are not on-screen, a multiple-instance (MI) (Maron & Lozano-Pérez, 1998) loss, which minimizes over the set of positive labels $\mathcal{R}$ may be more robust:

$$\mathcal{L}_{\mathrm{MI}}\left(y, \hat{y}\right) = \min_{m \in \mathcal{R}}\left(-\log\left(\hat{y}_m\right) - \sum_{m' \notin \mathcal{R}} \log\left(1 - \hat{y}_{m'}\right)\right). \tag{5}$$

**Active combinations cross entropy**: An alternative to the MI loss, *active combinations* (AC), corresponds to the minimum loss over all settings $\wp_{\geq 1}\left(\mathcal{R}\right)$ of the labels s.t. at least one label is 1:

$$\mathcal{L}_{\mathrm{AC}}\left(y, \hat{y}\right) = \min_{\mathcal{S} \in \wp_{\geq 1}(\mathcal{R})}\left(-\sum_{m \in \mathcal{S}} \log\left(\hat{y}_m\right) - \sum_{m' \notin \mathcal{S}} \log\left(1 - \hat{y}_{m'}\right)\right). \tag{6}$$

where $\wp_{\geq 1}\left(\mathcal{R}\right)$ denotes the power set of indices with label of 1.

## 4 EXPERIMENTAL FRAMEWORK

### 4.1 DATA PREPARATION

In order to train on real-world audio-visual recording environments for our open-domain system, we use the Yahoo Flickr Creative Commons 100 Million Dataset (YFCC100m) (Thomee et al., 2016). The dataset is drawn from about 200,000 videos (2,500 total hours) of various lengths and covering a diverse range of semantic sound categories. By splitting on video uploader, we select 1,600 videos for training, and use the remaining videos for validation and test. We extract 5-second clips with a hop size of 1 second, resulting in around 7.2 million clips. Clips consist of a 5-second audio waveform sampled at 16 kHz and 5 video frames $x^{(f)}$, where each frame is a $128 \times 128 \times 3$ RGB image.

Our goal is to train our system completely unsupervised, but we sought to reduce the proportion of videos with no on-screen sounds. We thus created a filtered subset $\mathcal{D}_f$ of YFCC100m of clips with a high audio-visual coincidence probability predicted by an unsupervised audio-visual coincidence prediction model (Jansen et al., 2020) trained on sounds from AudioSet (Gemmeke et al., 2017). The resulting selection is noisy, because the coincidence model is not perfect, and clips that have high audio-visual coincidence may contain both on-screen and off-screen sounds, or even no on-screen sounds. However, this selection does increase the occurrence of on-screen sounds, as shown below. The final filtered dataset consists of all clips (about 336,000) extracted from the 36,000 highest audio-visual coincidence scoring videos. The threshold for filtering was empirically set to keep a fair amount of diverse videos while ensuring that not too many off-screen-only clips were accepted.

To evaluate the performance of the unsupervised filtering and our proposed models, and to experiment with a small amount of supervised training data, we obtained human annotations for 10,000 unfiltered training clips, 10,000 filtered training clips, and 10,000 filtered validation/test clips. In the annotation process, the raters indicated "present" or "not present" for on-screen and off-screen sounds. Each clip is labeled by 3 individual raters, and is only considered on-screen-only or off-screen-only if raters are unanimous. We constructed an on-screen-only subset with 836 training, 735 validation, and 295 test clips, and an off-screen-only subset with 3,681 training, 836 validation, and 370 test clips.

Based on human annotations, we estimate that for unfiltered data 71.3% of clips contain both on-and-off-screen sounds, 2.8% contain on-screen-only sounds, and 25.9% only off-screen sounds. For the filtered data, 83.5% of clips contain on-screen and off-screen sounds, 5.6% of clips are on-screen-only, and 10.9% are off-screen-only. Thus, the unsupervised filtering reduced the proportion of off-screen-only clips and increased the proportion of clips with on-screen sounds.

## 4.2 TRAINING

Both audio and visual embedding networks were pre-trained on AudioSet (Gemmeke et al., 2017) for unsupervised coincidence prediction (Jansen et al., 2020) and fine-tuned on our data (see Appendix A.3.1 for ablation), whereas the separation network is trained from scratch using MixIT (2) on mixtures of mixtures (MoMs) from the audio of our data. All models are trained on 4 Google Cloud TPUs (16 chips) with Adam (Kingma & Ba, 2015), batch size 256, and learning rate $10^{-4}$.

To train the overall network, we construct minibatches of video clips, where the clip's audio is either a single video's soundtrack ("single mixture" example), or a mixture of two videos' soundtracks ("MoM" example): NOn (noisy-labeled on-screen), SOff (synthetic off-screen-only), LOn (human-labeled on-screen-only), and LOff (human-labeled off-screen-only). For all MoM examples, the second audio mixture is drawn from a different random video in the filtered data. Unsupervised minibatches consist of either 0% or 25% SOff examples, with the remainder as NOn. NOn examples are always MoMs, and SOff examples are evenly split between single mixtures and MoMs. A NOn MoM uses video clip frames and audio from the filtered high-coincidence subset of our data, $\mathcal{D}_f$, and SOff MoMs combine video frames of a filtered clip with random audio drawn from the dataset $\mathcal{D}_f$.

Semi-supervised minibatches additionally include LOn and LOff examples. Half of these examples in the minibatch are single-mixture examples, and the other half are MoM examples. LOn and LOff examples are constructed in the manner as NOn, except that the corresponding video clip is drawn from unanimously human-labeled on-screen-only videos and unanimously human-labeled off-screen-only videos, respectively. We experiment with using 0% or 25% SOff examples: (NOn, SOff) proportions of (50%, 0%) or (25%, 25%), respectively, with the remainder of the minibatch evenly split between LOn single-mixture, LOn MoM, LOff single-mixture, and LOff MoM.

Classification labels $y_m$ for all separated sources $\hat{s}_m$ in SOff and LOff examples are set to 0. For NOn and LOn examples, we set the label for each separated source as the first row of the MixIT mixing matrix (2): $y_m = A_{1,m}$. The MixIT separation loss (2) is used for all MoM example types.

## 4.3 EVALUATION

All evaluations use human-labeled test videos, which have been unanimously labeled as containing either only on-screen or only off-screen sounds. Using this data, we construct four evaluation sets: on-screen single mixtures, off-screen single mixtures, on-screen MoMs, and off-screen MoMs. The single-mixture evaluations consist of only data drawn from the particular label, either on-screen or

off-screen. Each on-screen (off-screen) MoM consists of an on-screen-only (off-screen-only) video clip, mixed with the audio from another random clip, drawn from the off-screen-only examples.

### 4.3.1 ON-SCREEN DETECTION

Detection performance for the on-screen classifier is measured using the area under the curve of the weighted receiver operator characteristic (AUC-ROC). Specifically, we set the weight for each source's prediction equal to the linear ratio of source power to input power, which helps avoid ambiguous classification decisions for inactive or very quiet sources. For single-mixture evaluations, positive labels are assigned for all separated sources from on-screen-only mixtures, and negative labels for all separated sources from off-screen-only mixtures. For on-screen MoM evaluations, labels for separated sources from on-screen MoMs are assigned using the first row of the oracle MixIT mixing matrix, and negative labels are assigned to sources separated from off-screen MoMs.

### 4.3.2 SEPARATION

Since we do not have access to individual ground-truth reference sources for our in-the-wild data, we cannot evaluate the per-source separation performance. The only references we have are mixtures. Thus, we compute an estimate of the on-screen audio by combining the separated sources using classifier predictions: $\hat{x}^{\text{on}} = \sum_{m=1}^{M} p_m \hat{s}_m$. For on-screen single mixture and MoM evaluations, we measure scale-invariant SNR (SI-SNR) (Le Roux et al., 2019), between $\hat{x}^{\text{on}}$ and the reference on-screen-only mixture $x^{(on)}$. SI-SNR measures the fidelity between a target $t \in \mathbb{R}^T$ and an estimate $\hat{t} \in \mathbb{R}^T$ within an arbitrary scale factor in units of decibels:

$$\text{SI-SNR}(t, \hat{t}) = 10 \log_{10} \frac{\|\alpha t\|^2}{\|\alpha t - \hat{t}\|^2}, \quad \alpha = \operatorname{argmin}_a \|at - \hat{t}\|^2 = \frac{t^T \hat{t}}{\|t\|^2}. \tag{7}$$

To measure the degree to which AudioScope rejects off-screen audio, we define the off-screen suppression ratio (OSR), which is the ratio in decibels of the power of the input mixture to the power of the on-screen estimate $\hat{x}^{\text{on}}$. We only compute OSR for off-screen evaluation examples where the input mixture only contains off-screen audio. Thus, higher OSR implies greater suppression of off-screen sounds. The minimum value of OSR is 0 dB, which means that $\hat{x}^{\text{on}}$ is equal to the input mixture, which corresponds to all on-screen classifier probabilities being equal to 1.

In some cases, SI-SNR and OSR might yield infinite values. For example, the estimate $\hat{y}$ may be zero, in which case SI-SNR (7) is $-\infty$ dB. This can occur when the input SNR of an on-screen mixture in a MoM is very low and none of the separated sources are assigned to it by MixIT. Conversely, if the estimate perfectly matches the target, SI-SNR can yield a value of $\infty$ dB, which occurs for on-screen single mixture evaluation cases when the separated sources trivially add up to the on-screen input due to mixture consistency of the separation model. For off-screen examples, OSR can also be infinite if the separation model achieves perfect off-screen suppression by predicting zero for $\hat{x}^{\text{on}}$. To avoid including these infinite values, we elect to measure median SI-SNR and OSR.

## 5 RESULTS

Results are shown in Table 1. Note that there is a trade-off between preservation of on-screen sounds, as measured by SI-SNR, and suppression of off-screen sounds, as measured by OSR: higher on-screen SI-SNR on on-screen examples generally means lower OSR on off-screen examples. Different classification losses have different operating points: for MoMs, compared to using the exact cross-entropy loss, models trained with active combinations or multiple instance loss achieve lower on-screen SI-SNR, while achieving more suppression (higher OSR) of off-screen sounds. Exact cross-entropy models achieve higher AUC-ROC for single mixtures and MoMs, and achieve better reconstruction of on-screen single mixtures at the expense of less rejection of off-screen mixtures.

Training only with the noisy labels provided by the unsupervised coincidence model (Jansen et al., 2020) achieves lower AUC-ROC compared to the semi-supervised condition that adds a small amount of human-labeled examples. Semi-supervised and unsupervised models achieve comparable on-screen SI-SNR, but semi-supervised models achieve better off-screen suppression. For example, the best on-screen SI-SNR for unsupervised and semi-supervised is 8.0 dB and 7.3 dB, respectively,

Table 1: Evaluation results for unanimously-annotated on-screen and off-screen mixtures. Training uses unsupervised or semi-supervised examples, with either $0\%$ or $25\%$ synthetic off-screen (SOff) examples (Section 4.2). Cross-entropy (CE) losses include active combinations (AC), multiple instance (MI), and exact. Note that "MixIT*" indicates SI-SNR of an oracle estimate derived using MixIT with reference mixtures, and $\hat{x}^{\mathrm{on}}$ is the on-screen estimate produced by mixing separated sources with classifier probabilities. On-screen MoMs have a median input SI-SNR of 4.4 dB.

| | | | | Single mixture | | | | Mixture of mixtures | | |
| | | | | On: SI-SNR (dB) | | Off: OSR (dB) | | On: SI-SNR (dB) | | Off: OSR (dB) |
| Training | SOff | CE loss | AUC | MixIT* | $\hat{x}^{\mathrm{on}}$ | $\hat{x}^{\mathrm{on}}$ | AUC | MixIT* | $\hat{x}^{\mathrm{on}}$ | $\hat{x}^{\mathrm{on}}$ |
|---|---|---|---|---|---|---|---|---|---|---|
| Unsup | 0% | AC | 0.58 | $\infty$ | 13.5 | 2.5 | 0.77 | 10.5 | 6.3 | 9.4 |
| Unsup | 0% | MI | 0.55 | $\infty$ | 11.9 | 3.6 | 0.75 | 10.1 | 5.3 | **10.8** |
| Unsup | 0% | Exact | 0.62 | $\infty$ | 36.6 | 0.5 | **0.81** | **10.6** | **8.0** | 5.3 |
| Unsup | 25% | AC | 0.57 | $\infty$ | 12.1 | **4.3** | 0.76 | 10.5 | 6.4 | 9.3 |
| Unsup | 25% | MI | 0.62 | $\infty$ | 13.6 | 4.0 | 0.78 | 9.6 | 6.1 | 9.6 |
| Unsup | 25% | Exact | **0.64** | $\infty$ | **41.0** | 0.6 | **0.81** | **10.6** | 7.5 | 4.5 |
| Semi | 0% | AC | 0.71 | $\infty$ | 14.8 | 6.6 | **0.82** | **10.4** | 6.1 | 14.1 |
| Semi | 0% | MI | 0.68 | $\infty$ | 12.3 | 11.3 | 0.79 | 9.6 | 4.7 | 21.0 |
| Semi | 0% | Exact | 0.73 | $\infty$ | **32.8** | 4.5 | 0.81 | 10.1 | **7.3** | 10.7 |
| Semi | 25% | AC | 0.79 | $\infty$ | 6.7 | **54.3** | 0.78 | 10.0 | 3.4 | **61.8** |
| Semi | 25% | MI | 0.82 | $\infty$ | 6.6 | 52.9 | 0.78 | 9.4 | 2.0 | 60.1 |
| Semi | 25% | Exact | **0.83** | $\infty$ | 6.6 | 53.9 | 0.81 | 10.0 | 2.4 | 61.5 |

while OSR is 5.3 dB and 10.7 dB. Using $25\%$ synthetic off-screen particularly shifts the behavior of semi-supervised models by biasing them towards predicting lower probabilities of on-screen. This bias results in lower on-screen SI-SNR, yet very strong off-screen rejection (i.e. very large OSRs).

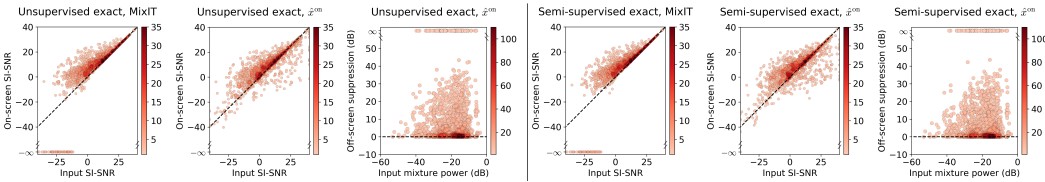

Figure 3: Scatter plots of input SI-SNR versus on-screen SI-SNR for on-screen evaluation examples, and mixture power in dB versus OSR. The settings for the models are SOff $0\%$ and exact CE loss, and colormap indicates density of points.

Figure 3 shows scatter plots of input SI-SNR versus SI-SNR of MixIT or $\hat{x}^{\mathrm{on}}$ on-screen estimates. From these plots, it is clear that the models tend to improve on-screen SI-SNR more often than not, and that these improvements are most significant around $\pm 10$ dB input SI-SNR. Note that for MixIT, a number of points have a SI-SNR of $-\infty$, which happens when MixIT assigns all separated sources to the off-screen mixture. OSR is sometimes $\infty$ when AudioScope achieves excellent off-screen suppression by predicting nearly 0 for the on-screen audio from off-screen-only input. To provide a sense of the qualitative performance of AudioScope, we include visualizations of best, worst, and typical predictions in the appendix, and the supplementary material contains audio-visual demos.

To benchmark AudioScope against other audio-visual separation approaches and measure performance on mismatched data, we evaluate on existing audio-visual separation test sets in Appendix A.2. We also performed a number of ablations for AudioScope, described in Appendix A.3.

## 6 CONCLUSION

In this paper we have proposed the first solution for training an unsupervised, open-domain, audio-visual on-screen separation system, without reliance on prior class labels or classifiers. We demonstrated the effectiveness of our system using a small amount of human-labeled, in-the-wild videos. A recipe for these will be available on the project webpage: `https://audioscope.github.io`. In future work, we will explore more fine-grained visual features, especially synchrony, which we expect will be especially helpful when multiple instances of the same object are present in the video. We also plan to use our trained classifier to refilter YFCC100m to get better noisy labels for the presence of on-screen sounds, which should further improve the performance of the system.

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

## A    APPENDIX

### A.1    QUALITATIVE EXAMPLES

For a range of input SNRs, Figure 4 shows best-case examples of separating on-screen sounds with AudioScope, while Figure 5 shows failure cases. Figures 6 and 7 show random examples at various SNRs, comparing the outputs of semi-supervised SOff $0\%$ models trained with either exact cross entropy (4) or active combinations cross entropy (6). Figure 6 shows the outputs of the two models on 7 random examples, and Figure 7 shows the outputs of the two models on 5 examples that have maximum absolute difference in terms of SI-SNR of the on-screen estimate.

The supplementary material includes audio-visual demos of AudioScope on single mixtures and MoMs with visualizations of MixIT assignments and predicted on-screen probabilities. For more examples please see: `https://audioscope.github.io/`.

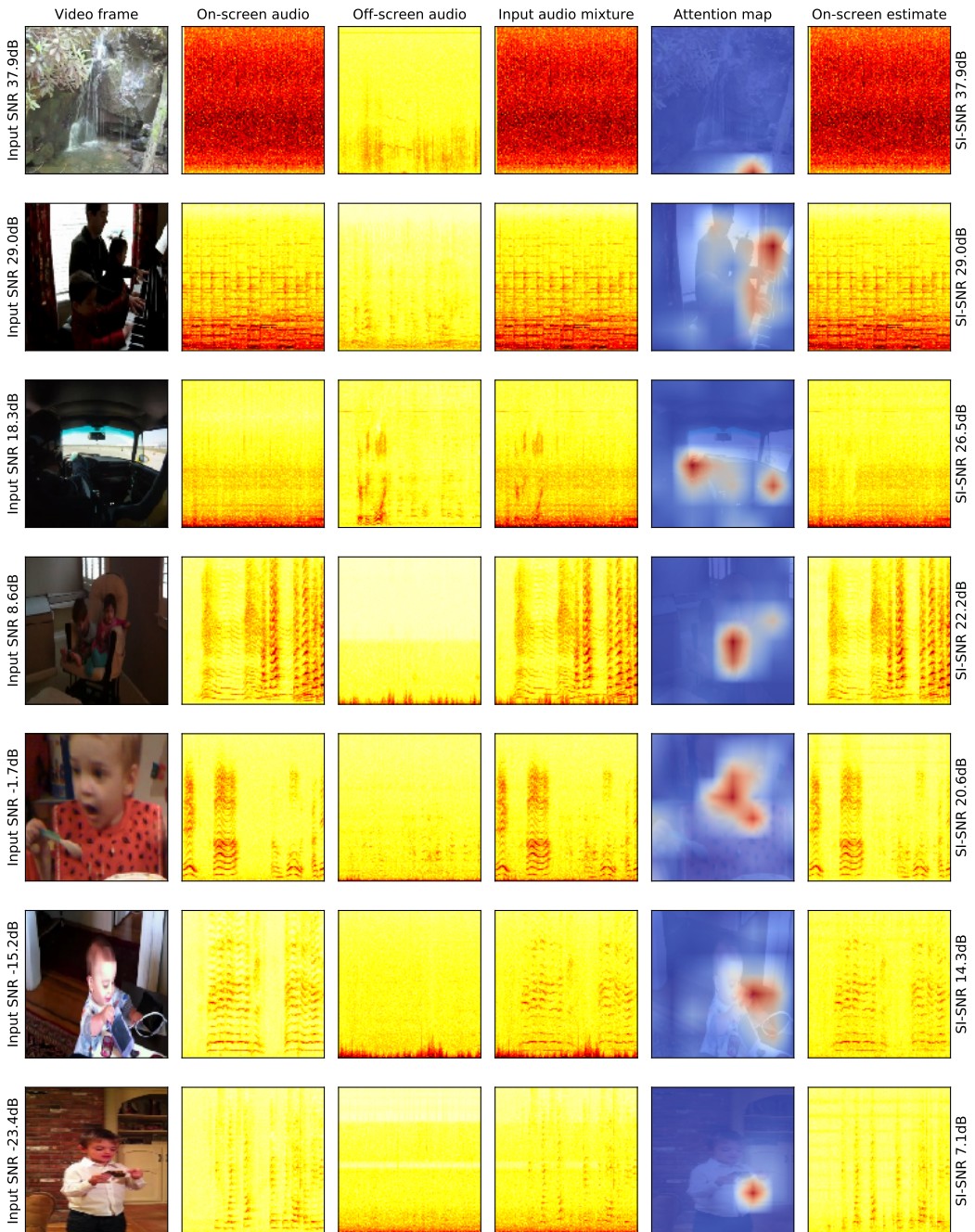

Figure 4: Best cases for AudioScope separation of on-screen sounds under various SNR conditions.

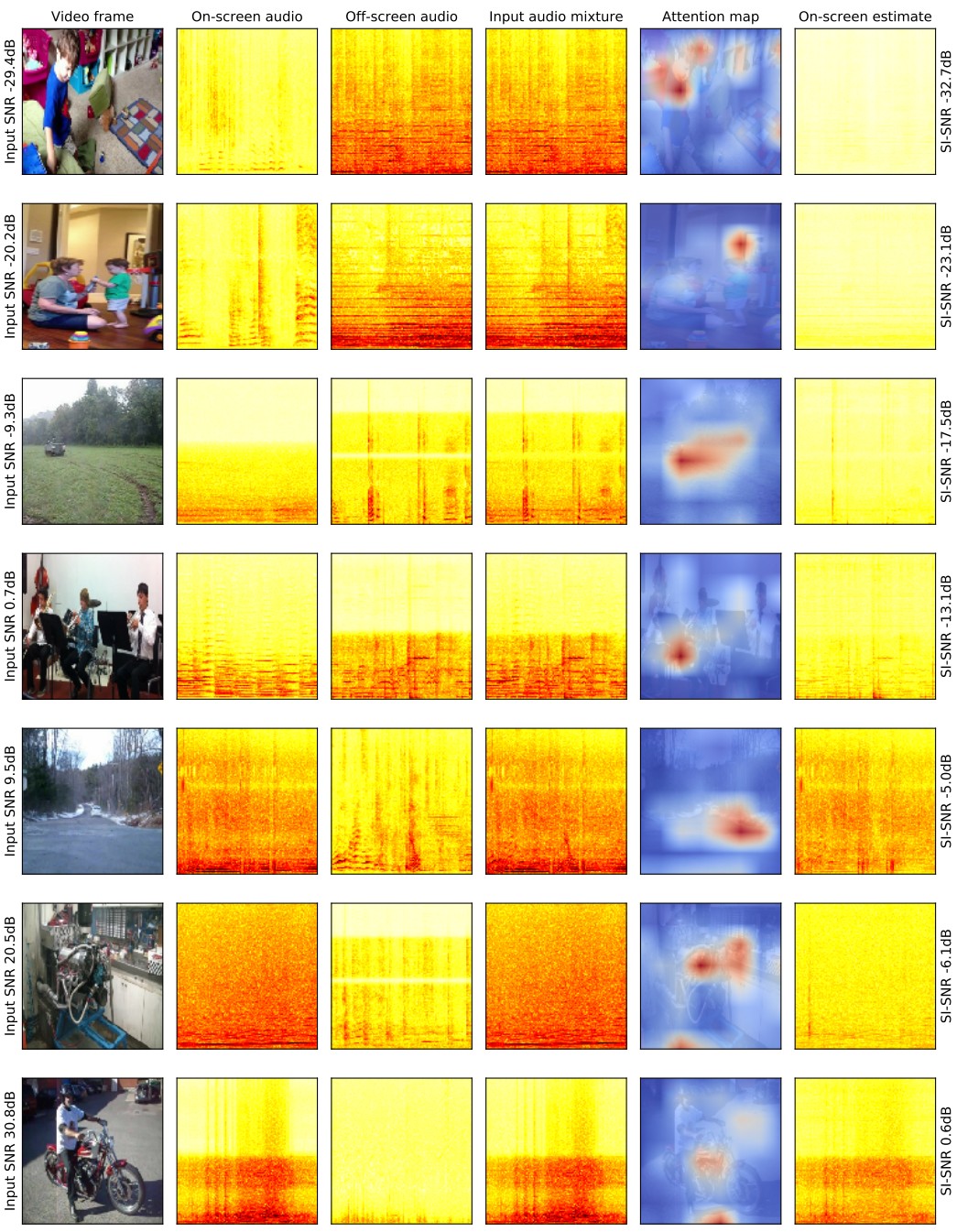

Figure 5: Failure cases for AudioScope separation of on-screen sounds under various SNR conditions.

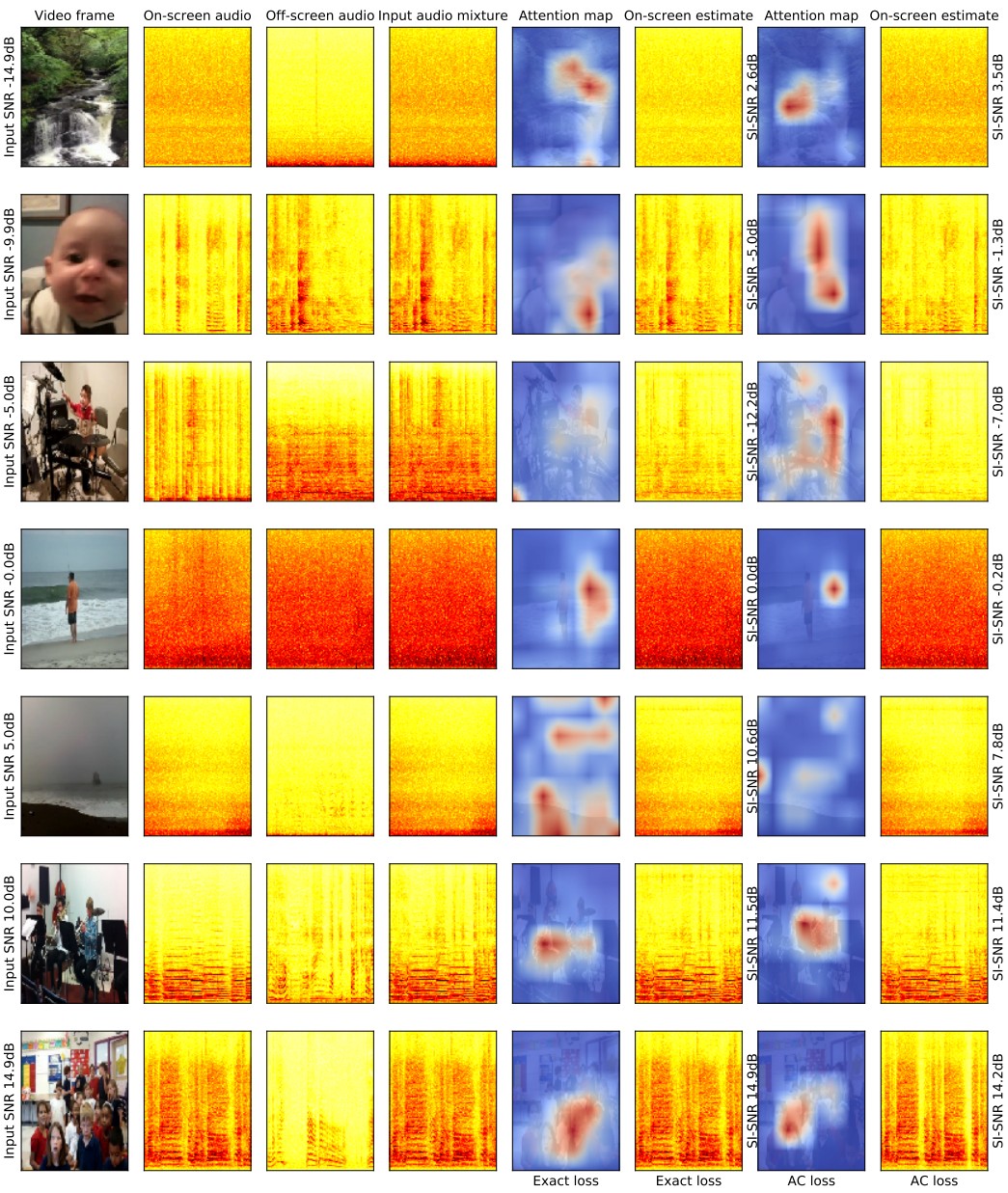

Figure 6: Comparison of random examples of separating on-screen sounds under various SNR conditions using either exact cross entropy (4) or active combinations cross entropy (6) as the classification loss.

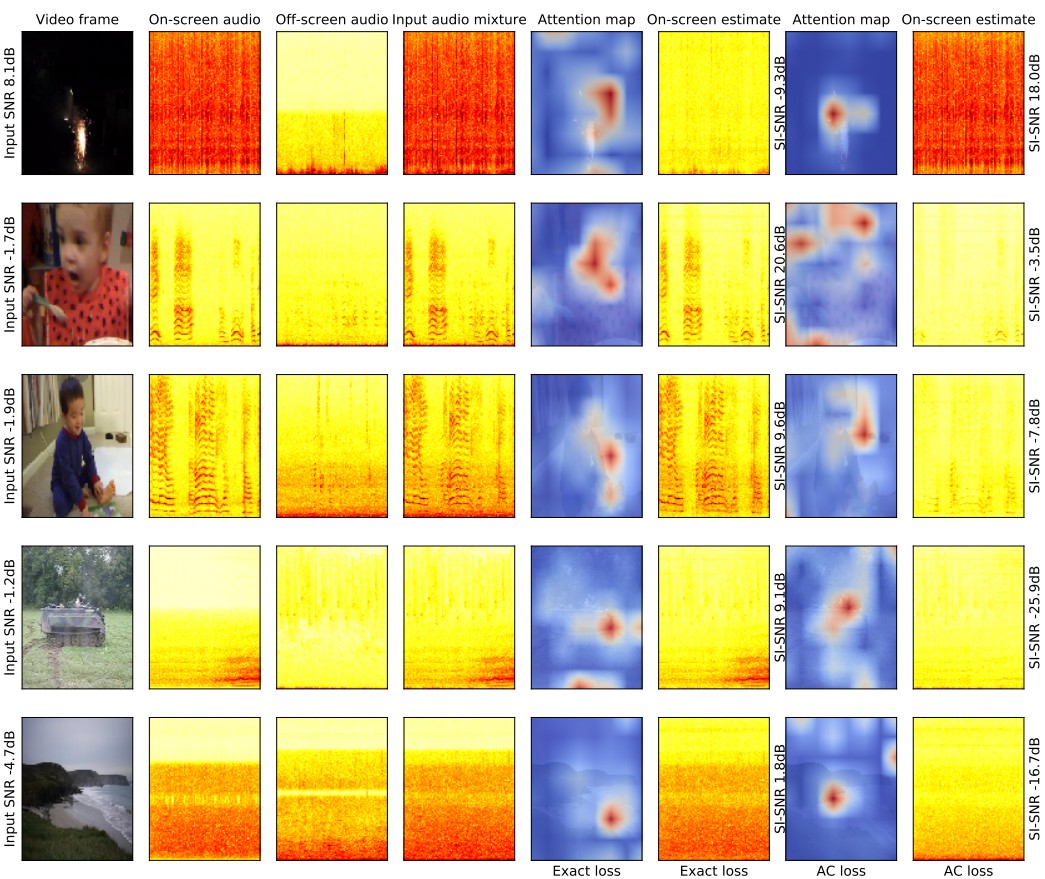

Figure 7: Comparison of examples with maximum absolute performance difference under various SNR conditions using either exact cross entropy (4) or active combinations cross entropy (6) as the classification loss.

## A.2 Evaluation on mismatched data

To evaluate the generalization capability of AudioScope and facilitate a comparison to prior works, we evaluated our model using test data from an audio-visual speech enhancement task (Hou et al., 2018) as well as an audio-visual task derived from a single-source subset of AudioSet (Gao et al., 2018). In both cases, the evaluation is on a restricted domain, and the prior methods used both matched and supervised training on that domain. In contrast, the AudioScope model is trained on open-domain YFCC100m videos using unsupervised training. For all evaluations we use the unsupervised AudioScope model using 0% SOff and active combinations loss.

### A.2.1 Evaluation on Audio-Visual Speech Enhancement

Since our method can be used to separate on-screen sounds for arbitrary classes of sound, to compare to existing approaches we evaluate the trained AudioScope model on the more restricted domain of audio-visual speech enhancement. To that end, we used the Mandarin sentences dataset, introduced by Hou et al. (2018). The dataset contains video utterances of Mandarin sentences spoken by a native speaker. Each sentence is unique and contains 10 Chinese characters. The length of each utterance is approximately 3 to 4 seconds. Synthetic noise is added to each ground truth audio. Forty such videos are used as the official testing set. For our evaluation we regard the speech of the filmed speaker as the on-screen sounds and the interference as off-screen sounds. Thus, we can compute quality metrics for the on-screen estimate while comparing to speech enhancement methods. To compare to previously-published numbers, we use signal-to-distortion ratio (SDR) (Vincent et al., 2006), which measures signal-to-noise ratio within a linear filtering of the reference signal.

Table 2 shows the comparison between Hou et al. (2018), Ephrat et al. (2018), AudioScope $\hat{x}^{\mathrm{on}}$ (on-screen estimate using predicted on-screen probabilities), AudioScope source with max $\hat{y}_m$ (use separated source with highest predicted on-screen probability), AudioScope best source (oracle selection of the separated source with the highest SDR with on-screen reference), and AudioScope MixIT* (on-screen estimate using oracle binary on-screen weights using references). Note that the AudioScope models are trained with mismatched open-domain training data, whereas the others were trained on matched speech enhancement data. It can be seen that although non-oracle AudioScope estimates do not advance on state-of-the-art performance of speech enhancement-specific methods, the oracle AudioScope estimates improve over (Hou et al., 2018). Thus AudioScope show promising results on this challenging data which is not explicitly represented in its open-domain training set. We believe that by adding such data to our training set, perhaps by fine-tuning, AudioScope could improve its performance significantly on this more specific task, which we leave for future work.

Table 2: Audio-visual enhancement results on the Mandarin test set (Hou et al., 2018).

| Method | In-Domain | Supervised | SDR | STOI |
|---|---|---|---|---|
| Hou et al. (2018) | ✓ | ✓ | 2.8 | 0.66 |
| Ephrat et al. (2018) | ✓ | ✓ | 6.1 | 0.71 |
| AudioScope $\hat{x}^{\mathrm{on}}$ | ✗ | ✗ | 2.5 | 0.59 |
| AudioScope source with max $\hat{y}_m$ | ✗ | ✗ | 2.3 | 0.58 |
| AudioScope best source (oracle) | ✗ | ✗ | 3.2 | 0.60 |
| AudioScope MixIT* (oracle) | ✗ | ✗ | 3.4 | 0.61 |

### A.2.2 Evaluation on AudioSet-SingleSource

We evaluated AudioScope on the musical instrument portion of the AudioSet-SingleSource dataset (Gao et al., 2018), which is a small number of clips from AudioSet (Gemmeke et al., 2017) that have been verified by humans to contain single sources. We use the same procedure as Gao & Grauman (2019) to construct a MoM test set, which creates 105 synthetic mixtures from all pairs of 15 musical instrument classes. For each pair, audio tracks are mixed together, and we perform separation twice for each pair, conditioning on the video for each source. The results are shown in table 3.

The non-oracle AudioScope methods perform rather poorly, but the oracle methods, especially MixIT* (which matches the MixIT training loss), achieve state-of-the-art performance compared to

Table 3: Audio-visual separation results on AudioSet-SingleSource musical instruments test set (Gao & Grauman, 2019).

| Method | In-Domain | Supervised | SDR | SIR | SAR |
|---|:---:|:---:|---|---|---|
| Sound-of-Pixels (Zhao et al., 2018) | ✓ | ✓ | 1.7 | 3.6 | 11.5 |
| AV-MIML (Gao et al., 2018) | ✓ | ✓ | 1.8 | - | - |
| Co-Separation (Gao & Grauman, 2019) | ✓ | ✓ | 4.3 | 7.1 | 13.0 |
| AudioScope $\hat{x}^{\mathrm{on}}$ | ✗ | ✗ | 0.4 | 2.7 | 11.4 |
| AudioScope source with max $\hat{y}_m$ | ✗ | ✗ | -0.9 | 2.8 | 7.9 |
| AudioScope best source (oracle) | ✗ | ✗ | 4.6 | 9.9 | 12.1 |
| AudioScope MixIT* (oracle) | ✗ | ✗ | 5.7 | 8.4 | 12.5 |

methods form the literature. This suggests that the on-screen classifier is less accurate on this data. Also, mixing the predicted AudioScope sources using the probabilities of the on-screen classifier may be suboptimal, and exploring alternative mixing methods to estimate on-screen audio is an avenue for future work. Fine-tuning on data for this specific task could also improve performance, which we also leave for future work.

### A.2.3 EVALUATION ON MUSIC

We also evaluated AudioScope on MUSIC (Zhao et al., 2018), which includes video clips of solo musical performances that have been verified by humans to contain single sources. We use the same procedure as Gao & Grauman (2019) to construct a MoM test set, which creates 550 synthetic mixtures from all 55 pairs of 11 musical instrument classes, with 10 random 10 second clips per pair[2]. For each pair, the two audio clips are mixed together, and we perform separation twice for each pair, conditioning on the video for each source. The results are shown in table 4.

Table 4: Audio-visual separation results on MUSIC test set (Zhao et al., 2018)

| Method | In-Domain | Supervised | SDR | SIR | SAR |
|---|:---:|:---:|---|---|---|
| Sound-of-Pixels (Zhao et al., 2018) | ✓ | ✓ | 5.4 | 11.0 | 9.8 |
| Sound-of-Motions (Zhao et al., 2019) | ✓ | ✓ | 4.8 | 11.0 | 8.7 |
| MP-Net Xu et al. (2019) | ✓ | ✓ | 5.7 | 11.4 | 10.4 |
| Co-Separation (Gao & Grauman, 2019) | ✓ | ✓ | 7.4 | 13.7 | 10.8 |
| Cascaded Opponent Filter (Zhu & Rahtu, 2020b) | ✓ | ✓ | 10.1 | 16.7 | 13.0 |
| A(Res-50, att) + S(DV3P) (Zhu & Rahtu, 2020a) | ✓ | ✓ | 9.4 | 15.6 | 12.7 |
| A(Res-50, class.) + S(DV3P) (Zhu & Rahtu, 2020a) | ✓ | ✓ | 10.6 | 17.2 | 12.8 |
| AudioScope $\hat{x}^{\mathrm{on}}$ | ✗ | ✗ | -0.5 | 2.8 | 11.2 |
| AudioScope source with max $\hat{y}_m$ | ✗ | ✗ | -2.0 | 3.3 | 7.6 |
| AudioScope best source (oracle) | ✗ | ✗ | 7.1 | 14.9 | 12.5 |
| AudioScope MixIT* (oracle) | ✗ | ✗ | 8.8 | 13.0 | 13.1 |

We see a similar pattern compared to the results for AudioSet-SingleSource in Table 3: non-oracle methods that use the predicted on-screen probability $\hat{y}_m$ do not perform very well. However, oracle selection of the best source, or oracle remixing of the sources, both achieve better performance than a number of recent specialized supervised in-domain systems from the literature, though they do not achieve state-of-the-art performance. These results seem to suggest that the predictions $\hat{y}_m$ are less accurate for this restricted-domain task, but the excellent oracle results suggest potential. In particular, non-oracle performance could improve if the classifier were more accurate, perhaps by fine-tuning. Also, there may be better ways of combining separated sources together to reconstruct on-screen sounds.

---

[2] Gao & Grauman (2019) did not provide the exact clip timestamps they used. We used a sliding window of 10 seconds with a hop of 5 seconds, and randomly selected 10 of these.

## A.3 ABLATIONS

We performed a number of ablations on AudioScope. The following subsections show the results of a number of ablations using either unsupervised or semi-supervised training. All models for these ablation use 0% SOff examples and the active combinations loss (6).

### A.3.1 AUDIO AND VIDEO EMBEDDINGS

Table 5 shows the results of various ablations involving audio and video embeddings in the model.

Table 5: Ablations related to audio and video embeddings.

| | | Single mixture | | | Mixture of mixtures | | | |
| | | | On: SI-SNR | Off: OSR | | On: SI-SNR | | Off: OSR |
| Training | Ablation | AUC | $\hat{x}^{\mathrm{on}}$ | $\hat{x}^{\mathrm{on}}$ | AUC | MixIT* | $\hat{x}^{\mathrm{on}}$ | $\hat{x}^{\mathrm{on}}$ |
|---|---|---|---|---|---|---|---|---|
| Unsup | – | 0.58 | 13.5 | 2.5 | 0.77 | 10.5 | 6.3 | 9.4 |
| Unsup | No video conditioning for separation | 0.54 | 11.5 | 2.7 | 0.75 | 10.6 | 5.4 | 11.2 |
| Unsup | Emb. networks from scratch | 0.45 | 8.9 | 7.0 | 0.61 | 10.6 | 2.4 | 13.4 |
| Unsup | No global video and audio emb. to classifier | 0.55 | 14.0 | 1.1 | 0.77 | 10.3 | 6.9 | 5.9 |
| Unsup | No global video emb. to classifier | 0.57 | 13.3 | 2.3 | 0.77 | 10.6 | 6.3 | 9.8 |
| Semi | – | 0.71 | 14.8 | 6.6 | 0.82 | 10.4 | 6.1 | 14.1 |
| Semi | No video conditioning for separation | 0.65 | 12.8 | 12.3 | 0.77 | 10.4 | 5.3 | 19.7 |
| Semi | Emb. networks from scratch | 0.48 | 6.3 | 12.7 | 0.59 | 9.7 | 2.0 | 16.4 |
| Semi | No global video and audio emb. to classifier | 0.69 | 11.1 | 14.7 | 0.78 | 10.1 | 4.5 | 21.4 |
| Semi | No global video emb. to classifier | 0.66 | 11.7 | 10.0 | 0.77 | 10.2 | 5.6 | 16.3 |

First, notice that removing video conditioning for the separation model reduces on-screen SI-SNR by 2 dB on single mixtures and 0.9 dB on MoMs, with negligible or slight improvement in OSR. Thus, we can conclude that visual conditioning does have some benefit for the model.

Next, we consider training the audio and video embedding networks from scratch, instead of using the coincidence model weights pretrained using AudioSet (Jansen et al., 2020). Training from scratch is quite detrimental, as AUC-ROC decreases by a minimum of 0.13 and maximum of 0.23 across single-mixtures/MoMs and unsupervised/semi-supervised conditions. Furthermore, separation performance suffers, with on-screen SI-SNR dropping by multiple for all conditions.

Finally, we consider removing the global video embedding, or both the global video embedding and audio embeddings, from the input of the on-screen classifier. This results in equivalent or slightly worse AUC-ROC, with equivalent or worse on-screen SI-SNR. For unsupervised training, removing both embeddings at the classifier input improves on-screen SI-SNR a bit (0.5 dB for single mixtures, 0.6 dB for MoMs) with a slight drop in OSR, though for semi-supervised on-screen SI-SNR drops by 3.7 dB for single mixtures and 0.5 dB for MoMs. Overall, the best result is achieved by including these embeddings at the classifier input.

### A.3.2 ATTENTIONAL POOLING

We tried decreasing the embedding dimension from 256 to 128, as well as replacing the attentional pooling with mean pooling for audio sources, video frames, or both. The results are shown in Table 6.

Table 6: Ablations related to attentional pooling run for the unsupervised setting.

| | | Single mixture | | | Mixture of mixtures | | | |
| | | | On: SI-SNR | Off: OSR | | On: SI-SNR | | Off: OSR |
| Training | Ablation | AUC | $\hat{x}^{\mathrm{on}}$ | $\hat{x}^{\mathrm{on}}$ | AUC | MixIT* | $\hat{x}^{\mathrm{on}}$ | $\hat{x}^{\mathrm{on}}$ |
|---|---|---|---|---|---|---|---|---|
| Unsup | – | 0.58 | 13.5 | 2.5 | 0.77 | 10.5 | 6.3 | 9.4 |
| Unsup | Attention embedding dimension 128 | 0.56 | 12.1 | 1.9 | 0.76 | 10.1 | 5.7 | 7.5 |
| Unsup | No attentional pooling for audio sources | 0.59 | 14.3 | 1.9 | 0.78 | 10.3 | 6.4 | 7.5 |
| Unsup | No attentional pooling for video | 0.57 | 16.2 | 1.4 | 0.77 | 10.5 | 6.8 | 6.2 |
| Unsup | No attentional pooling for audio and video | 0.56 | 12.8 | 1.8 | 0.76 | 10.5 | 6.3 | 7.7 |

Decreasing the embedding dimension reduces performance, dropping on-screen SI-SNR by 1.4 dB on single mixtures and 0.6 dB on MoMs, also with reduction in OSR. Replacing attentional pooling with mean pooling generally does not change AUC-ROC or on-screen SI-SNR that much, but does result

in a OSR reduction of at least 0.6 dB for single mixtures and 1.7 dB for MoMs. Thus, attentional pooling seems to have a beneficial effect in that it improves off-screen suppression, with equivalent classification and on-screen separation performance.

### A.3.3 DATA FILTERING

As described in section 4.1, we use an unsupervised audio-visual coincidence model (Jansen et al., 2020) to filter training videos for on-screen sounds. To ablate the benefit of this filtering, we tried using different combinations of filtered and unfiltered data for NOn examples, as described in section 4.2, which uses filterd data for both on-screen and off-screen mixtures. Filtered data has the advantage of less noisy on-screen labels, but the disadvantage that it lacks the variety of unfiltered data, being only 4.7% of the unfiltered data.

Table 7: Ablations for different data configurations.

| | | Single mixture | | | Mixture of mixtures | | | |
| | | | On: SI-SNR | Off: OSR | | On: SI-SNR | | Off: OSR |
| Training | Ablation | AUC | $\hat{x}^{on}$ | $\hat{x}^{on}$ | AUC | MixIT* | $\hat{x}^{on}$ | $\hat{x}^{on}$ |
|---|---|---|---|---|---|---|---|---|
| Unsup | – | 0.58 | 13.5 | 2.5 | 0.77 | 10.5 | 6.3 | 9.4 |
| Unsup | Filtered on-screen, unfiltered off-screen | 0.60 | 11.5 | 3.4 | 0.78 | 10.7 | 5.7 | 11.0 |
| Unsup | Unfiltered on-screen, unfiltered off-screen | 0.60 | 12.6 | 2.5 | 0.77 | 10.5 | 6.4 | 6.7 |
| Unsup | Unfiltered on-screen, filtered off-screen | 0.63 | 15.3 | 2.3 | 0.79 | 10.7 | 7.0 | 5.6 |
| Semi | – | 0.71 | 14.8 | 6.6 | 0.82 | 10.4 | 6.1 | 14.1 |
| Semi | Filtered on-screen, unfiltered off-screen | 0.68 | 11.1 | 7.1 | 0.78 | 10.2 | 5.3 | 13.1 |
| Semi | Unfiltered on-screen, unfiltered off-screen | 0.66 | 13.5 | 6.0 | 0.76 | 10.4 | 6.2 | 9.6 |
| Semi | Unfiltered on-screen, filtered off-screen | 0.65 | 14.8 | 7.8 | 0.74 | 10.4 | 5.6 | 8.9 |

The results are shown in Table 7. For unsupervised training, unfiltered on-screen with filtered off-screen achieves improved performance in terms of AUC-ROC and on-screen SI-SNR, yet OSR decreases for MoMs. This suggests that in the absence of cleanly-labeled on-screen videos, a larger amount of data with noisier labels is better compares to a smaller amount of data with less noisy labels. However, for semi-supervised training that includes a small amount of cleanly-labeled on-screen examples, AUC-ROC is consistently worse for all ablations, and on-screen SI-SNR and OSR are generally equivalent or worse for all ablations. Thus, these ablations validate that using filtered data for both on-screen and off-screen components of NOn examples with semi-supervised training achieves the best results overall.

### A.3.4 NUMBER OF OUTPUT SOURCES

For all experiments in this paper, we generally used $M = 4$ output sources for the separation model, which is the maximum number of sources that it can predict. Here we see if increasing the number of output sources can improve performance. More output source slots provides a separation model with more flexibility in decomposing the input waveform, yet the drawback is that the model may over-separate (i.e. split sources into multiple components), and there is more pressure on the classifier to correctly group components of the on-screen sound together. The results are shown in Table 8.

Table 8: Ablations for number of max output sources.

| | | Single mixture | | | Mixture of mixtures | | | |
| | | | On: SI-SNR | Off: OSR | | On: SI-SNR | | Off: OSR |
| Training | Ablation | AUC | $\hat{x}^{on}$ | $\hat{x}^{on}$ | AUC | MixIT* | $\hat{x}^{on}$ | $\hat{x}^{on}$ |
|---|---|---|---|---|---|---|---|---|
| Unsup | – | 0.58 | 13.5 | 2.5 | 0.77 | 10.5 | 6.3 | 9.4 |
| Unsup | 6 output sources | 0.54 | 9.6 | 2.1 | 0.71 | 10.3 | 5.0 | 6.7 |
| Unsup | 8 output sources | 0.52 | 6.9 | 4.5 | 0.69 | 11.1 | 3.6 | 10.1 |
| Semi | – | 0.71 | 14.8 | 6.6 | 0.82 | 10.4 | 6.1 | 14.1 |
| Semi | 6 output sources | 0.67 | 7.9 | 12.4 | 0.76 | 10.8 | 4.0 | 20.4 |
| Semi | 8 output sources | 0.67 | 6.5 | 18.8 | 0.77 | 10.9 | 2.6 | 24.6 |

For unsupervised training, increasing the number of output sources generally degrades AUC-ROC and on-screen SI-SNR, while boosting OSR a bit. Note that the MixIT* improves for MoMs with 8 output sources (10.5 dB → 11.1 dB), which suggests the greater flexibility of the model, yet the

on-screen estimate $\hat{x}^{\mathrm{on}}$ is quite a bit worse (3.6 dB), also compared to on-screen SI-SNR for 4 output sources (6.3 dB).

For semi-supervised training, MixIT* performance also improves with more output sources, but AUC-ROC and on-screen SI-SNR decrease, suggesting the increased pressure on the classifier to make correct predictions for more, and potentially partial, sources. OSR increases with more output sources, which suggests the classifier biases towards predicting 0s more often. Thus, increasing the number of sources shifts the operating point of the model away from separating on-screen sounds and towards suppressing off-screen sounds.

### A.3.5 BASELINE SEPARATION MODEL

We also trained two-output baseline separation models without the on-screen classifier, where the first estimated source is the on-screen estimate $\hat{x}^{\mathrm{on}}$ with training target of on-screen audio, and the second estimated source is the off-screen estimate $\hat{x}^{\mathrm{off}}$ with training target of off-screen audio. These models were trained with or without video conditioning, using the negative SNR loss (3). The training data is exactly the same as in Table 1, with 0% SOff.

Table 9: Results for baseline two-output separation model without on-screen classifier.

| | | Single mixture | | | | |
| | | On-screen | | | Off-screen | |
| Training | Video cond. | SI-SNR $\hat{x}^{\mathrm{on}}$ | ISR $\hat{x}^{\mathrm{on}}$ | ISR $\hat{x}^{\mathrm{off}}$ | ISR $\hat{x}^{\mathrm{on}}$ | ISR $\hat{x}^{\mathrm{off}}$ |
|---|---|---|---|---|---|---|
| Predict input for $\hat{x}^{\mathrm{on}}$ and 0 for $\hat{x}^{\mathrm{off}}$ | | $\infty$ | 0.0 | $\infty$ | 0.0 | $\infty$ |
| Predict $1/2$ input for $\hat{x}^{\mathrm{on}}$ and $\hat{x}^{\mathrm{off}}$ | | $\infty$ | 3.0 | 3.0 | 3.0 | 3.0 |
| Predict 0 for $\hat{x}^{\mathrm{on}}$ and input for $\hat{x}^{\mathrm{off}}$ | | NaN | $\infty$ | 0.0 | $\infty$ | 0.0 |
| Unsup | ✗ | 66.2 | 6.0 | 6.0 | 6.0 | 6.0 |
| Unsup | ✓ | 29.7 | 0.2 | 27.3 | 2.4 | 6.7 |
| Semi | ✗ | -18.0 | 51.1 | 0.0 | 51.2 | 0.0 |
| Semi | ✓ | 18.8 | 10.5 | 3.0 | 48.5 | 0.0 |
| | | Mixture of mixtures | | | | |
| | | On-screen | | | Off-screen | |
| Training | Video cond. | SI-SNR $\hat{x}^{\mathrm{on}}$ | ISR $\hat{x}^{\mathrm{on}}$ | ISR $\hat{x}^{\mathrm{off}}$ | ISR $\hat{x}^{\mathrm{on}}$ | ISR $\hat{x}^{\mathrm{off}}$ |
| Predict input for $\hat{x}^{\mathrm{on}}$ and 0 for $\hat{x}^{\mathrm{off}}$ | | 4.4 | 0.0 | $\infty$ | 0.0 | $\infty$ |
| Predict $1/2$ input for $\hat{x}^{\mathrm{on}}$ and $\hat{x}^{\mathrm{off}}$ | | 4.4 | 3.0 | 3.0 | 3.0 | 3.0 |
| Predict 0 for $\hat{x}^{\mathrm{on}}$ and input for $\hat{x}^{\mathrm{off}}$ | | NaN | $\infty$ | 0.0 | $\infty$ | 0.0 |
| Unsup | ✗ | 4.4 | 6.0 | 6.0 | 6.0 | 6.0 |
| Unsup | ✓ | 4.1 | 5.8 | 3.7 | 9.5 | 1.7 |
| Semi | ✗ | -19.7 | 53.3 | 0.0 | 54.2 | 0.0 |
| Semi | ✓ | -5.3 | 29.5 | 0.3 | 53.3 | 0.0 |

Table 9 shows the results in terms of the same metrics used in Tables 1, except that instead of "off-screen rejection ratio (OSR)", we report "input-to-source ratio (ISR)" (i.e. $10 \log_{10}$ of the ratio of input power to estimated source power) for each of the two output sources. High ISR means that the source power is lower compared to the input power. Note that ISR $\hat{x}^{\mathrm{on}}$ for off-screen single-mixtures and MoMs is equivalent to OSR. Table 9 also includes several trivial baselines with expected scores.

First, notice that none of these models approach the performance of separation models that include the on-screen classifier, as shown in Table 1. Second, the unsupervised and semi-supervised models here achieve distinctly different operating points. Without video conditioning, the unsupervised model achieves a trivial solution, nearly equivalent to just outputting $1/2$ the input mixture for each estimated source. Adding video conditioning for the unsupervised model actually reduces single-mixture performance a bit (66.2 dB to 29.7 dB).

The semi-supervised model without video conditioning is very poor at single-mixture on-screen SI-SNR (-18.0 dB), yet achieves quite high single-mixture OSR (51.1 dB). As indicated by the ISRs,

the model tends to prefer nearly-zero on-screen estimates, which may be due to the additional cleanly-labeled off-screen examples provided during training. For the video-conditioned semi-supervised model, single-mixture on-screen SI-SNR improves by quite a lot (-18.0 dB to 18.8 dB), but on-screen SI-SNR performance for on-screen MoMs is abysmal (-19.7 dB without visual conditioning, -5.3 dB with visual conditioning).

Overall, we can conclude from these baselines that simply training a two-output separation model with on-screen and off-screen targets, even with visual conditioning, is not a feasible approach for our open-domain and noisily-labeled data.

## A.4 Neural network architectures

We briefly present the architectures used in this work for the separation network $\mathcal{M}^{\mathrm{s}}$, the audio embedding network $\mathcal{M}^{\mathrm{a}}$, and the image embedding network $\mathcal{M}^{\mathrm{v}}$, and referred in Sections 3.1, 3.2 and 3.3, respectively.

We present the architecture of the TDCN++ separation network in Table 10. The input to the separation network is a mixture waveform with $T$ time samples and the output is a tensor containing the $M$ estimated source waveforms $\hat{s} \in \mathbb{R}^{M \times T}$. The input for the $i$th depth-wise (DW) separable convolutional block is the summation of all skip-residual connections and the output of the previous block. Specifically, there are the following skip connections defined w.r.t. their index $i = 0, \ldots, 31$: $0 \to 8, 0 \to 16, 0 \to 24, 8 \to 16, 8 \to 24$ and $16 \to 24$.

Table 10: TDCN++ separation network architecture for an input mixture waveform corresponding to a time-length of 5 seconds, sampled at 16 kHz. The output of the separation network are $M = 4$ separated sources. The dilation factor for each block is defined as $D_i = 2^{\mathrm{mod}(i,8)}, i = 0, \ldots, 31$.

| Multiples | Layer operation | Filter size | Stride | Dilation | Input shape |
|---|---|---|---|---|---|
| ×1 | Conv1D | $40 \times 1 \times 256$ | 20 | 1 | $1 \times 80,000$ |
| | Dense | $1 \times 256 \times 256$ | 1 | 1 | $256 \times 4,000$ |
| ×32 | Dense | $1 \times 256 \times 512$ | 1 | 1 | $256 \times 4,000$ |
| | PReLU | $1 \times 1 \times 1$ | – | – | $512 \times 4,000$ |
| | Instance Norm | $1 \times 512$ Separable | – | – | $512 \times 4,000$ |
| | DW Conv1D | $3 \times 512$ Separable | 1 | $D_i$ | $512 \times 4,000$ |
| | PReLU | $1 \times 1 \times 1$ | – | – | $512 \times 4,000$ |
| | Instance Norm | $1 \times 512$ Separable | – | – | $512 \times 4,000$ |
| | Dense | $1 \times 512 \times 256$ | 1 | 1 | $512 \times 4,000$ |
| ×1 | Dense | $1 \times 256 \times M \cdot 256$ | 1 | 1 | $256 \times 4,000$ |
| | Reshape | – | – | – | $M \cdot 256 \times 4,000$ |
| | ConvTranspose1D | $40 \times 256 \times 1$ | 20 | 1 | $M \times 256 \times 4,000$ |

In a similar way, in Table 11 we define the image and audio embedding networks, which use the same MobileNet v1 architecture (Howard et al., 2017) with different input tensors.

The extraction of each image embedding $Z_j^{\mathrm{v}}$, $j = 1, \ldots, 5$ relies on the application of the image embedding network $\mathcal{M}^{\mathrm{v}}$ on top of each input video frame individually. Moreover, in order to extract the local video spatio-temporal embedding, we extract the output of the $8 \times 8$ convolutional map (denoted with a * in Table 11) for each input video frame and feed it through a dense layer in order to reduce its channel dimensions to 1. By concatenating all these intermediate convolutional maps we form the local spatio-temporal video embedding $Z^{\mathrm{vl}}$ as specified in Section 3.3.

On the other hand, we extract a time-varying embedding $Z_m^{\mathrm{a}}$ for the $m$th separated source waveform by applying the audio embedding network $\mathcal{M}^{\mathrm{a}}$ on overlapping audio segments and concatenating those outputs. The audio segments are extracted with an overlap of 86 windows or equivalently 0.86 seconds. Specifically, for each segment, we extract the mel-spectrogram representation from 96 windows with a length of 25ms and a hop size of 10ms forming the input for the audio embedding network as a matrix with size $96 \times 64$, where 64 is the number of mel-features. After feeding this mel-spectrogram as an input to our audio embedding network $\mathcal{M}^{\mathrm{a}}$, we extract the corresponding static length representation for this segment $Z_j^{\mathrm{a}}$, where $j$ denotes the segment index.

Table 11: Audio and image embedding network architectures for an input segment log-mel spectrogram corresponding to a 0.96 seconds, sampled at 16 kHz and an input image represented as an RGB tensor with shape $128 \times 128 \times 3$, respectively. The log-mel spectrogram input to the audio embedding network has a number of input channels $C_{in} = 1$ and the input video frame to the image embedding network has a number of input channels $C_{in} = 3$. Each depth-wise (DW) convolution or regular convolution is followed by a batch normalization layer and a ReLU activation. * denotes the layer from which the local $8 \times 8$ spatial feature map is extracted, as described in Section 3.3.

| Layer operation | Filter size | Stride | Input shape | |
|---|---|---|---|---|
| | | | Audio network | Image network |
| Conv | $3 \times 3 \times C_{in} \times 32$ | 2 | $96 \times 64 \times 1$ | $128 \times 128 \times 3$ |
| DW Conv | $3 \times 3 \times 32$ Separable | 1 | $48 \times 32 \times 32$ | $64 \times 64 \times 32$ |
| Conv | $1 \times 1 \times 32 \times 64$ | 1 | $48 \times 32 \times 32$ | $64 \times 64 \times 32$ |
| DW Conv | $3 \times 3 \times 64$ Separable | 2 | $48 \times 32 \times 64$ | $64 \times 64 \times 64$ |
| Conv | $1 \times 1 \times 64 \times 128$ | 1 | $24 \times 16 \times 64$ | $32 \times 32 \times 64$ |
| DW Conv | $3 \times 3 \times 128$ Separable | 1 | $24 \times 16 \times 128$ | $32 \times 32 \times 128$ |
| Conv | $1 \times 1 \times 128 \times 128$ | 1 | $24 \times 16 \times 128$ | $32 \times 32 \times 128$ |
| DW Conv | $3 \times 3 \times 128$ Separable | 2 | $24 \times 16 \times 128$ | $32 \times 32 \times 128$ |
| Conv | $1 \times 1 \times 128 \times 256$ | 1 | $12 \times 8 \times 128$ | $16 \times 16 \times 128$ |
| DW Conv | $3 \times 3 \times 256$ Separable | 1 | $12 \times 8 \times 256$ | $16 \times 16 \times 256$ |
| Conv | $1 \times 1 \times 256 \times 256$ | 1 | $12 \times 8 \times 256$ | $16 \times 16 \times 256$ |
| DW Conv | $3 \times 3 \times 256$ Separable | 2 | $12 \times 8 \times 256$ | $16 \times 16 \times 256$ |
| Conv | $1 \times 1 \times 256 \times 512$ | 1 | $6 \times 4 \times 256$ | $8 \times 8 \times 256$ |
| DW Conv | $3 \times 3 \times 512$ Separable | 1 | $6 \times 4 \times 512$ | $8 \times 8 \times 512$ |
| Conv * | $1 \times 1 \times 512 \times 512$ | 1 | $6 \times 4 \times 512$ | $8 \times 8 \times 512$ |
| DW Conv | $3 \times 3 \times 512$ Separable | 1 | $6 \times 4 \times 512$ | $8 \times 8 \times 512$ |
| Conv | $1 \times 1 \times 512 \times 512$ | 1 | $6 \times 4 \times 512$ | $8 \times 8 \times 512$ |
| DW Conv | $3 \times 3 \times 512$ Separable | 1 | $6 \times 4 \times 512$ | $8 \times 8 \times 512$ |
| Conv | $1 \times 1 \times 512 \times 512$ | 1 | $6 \times 4 \times 512$ | $8 \times 8 \times 512$ |
| DW Conv | $3 \times 3 \times 512$ Separable | 1 | $6 \times 4 \times 512$ | $8 \times 8 \times 512$ |
| Conv | $1 \times 1 \times 512 \times 512$ | 1 | $6 \times 4 \times 512$ | $8 \times 8 \times 512$ |
| DW Conv | $3 \times 3 \times 512$ Separable | 1 | $6 \times 4 \times 512$ | $8 \times 8 \times 512$ |
| Conv | $1 \times 1 \times 512 \times 512$ | 1 | $6 \times 4 \times 512$ | $8 \times 8 \times 512$ |
| DW Conv | $3 \times 3 \times 512$ Separable | 1 | $6 \times 4 \times 512$ | $8 \times 8 \times 512$ |
| Conv | $1 \times 1 \times 512 \times 512$ | 1 | $6 \times 4 \times 512$ | $8 \times 8 \times 512$ |
| DW Conv | $3 \times 3 \times 512$ Separable | 2 | $6 \times 4 \times 512$ | $8 \times 8 \times 512$ |
| Conv | $1 \times 1 \times 512 \times 1024$ | 1 | $3 \times 2 \times 512$ | $4 \times 4 \times 512$ |
| DW Conv | $3 \times 3 \times 1024$ Separable | 1 | $3 \times 2 \times 1024$ | $4 \times 4 \times 1024$ |
| Conv | $1 \times 1 \times 1024 \times 1024$ | 1 | $3 \times 2 \times 1024$ | $4 \times 4 \times 1024$ |
| Average Pooling | – | – | $3 \times 2 \times 1024$ | $4 \times 4 \times 1024$ |
| Dense | $1 \times 1 \times 1024 \times 128$ | 1 | $1 \times 1 \times 1024$ | $1 \times 1 \times 1024$ |

### A.5 HUMAN EVALUATION

To determine the subjective quality of AudioScope predictions, we performed another round of human annotation on on-screen test MoM videos. The rating task is the same as the one used to annotate data, as described in Section 4.1, where raters were asked to mark the presence of on-screen sounds and off-screen sounds. All models for these evaluations are the same as the base model used in Appendix A.3: 0% SOff examples with active combinations loss (6). Each example was annotated by 3 raters, and the ultimate binary rating for each example is determined by majority.

Table 12: Results of human annotation task on on-screen MoM test set.

| Method | % on-screen only | % off-screen only | % on-and-off-screen | % unsure |
|---|---|---|---|---|
| Unprocessed | 24.1 | 2.7 | 67.4 | 5.9 |
| Unsup $\hat{x}^{\mathrm{on}}$ | 38.0 | 2.5 | 53.8 | 5.7 |
| Unsup MixIT* | 37.1 | 1.1 | 53.7 | 8.0 |
| Semisup $\hat{x}^{\mathrm{on}}$ | 37.6 | 2.3 | 54.0 | 6.1 |
| Semisup MixIT* | 36.9 | 1.2 | 52.8 | 9.1 |

The results for the on-screen MoM test set are shown in Table 12. We evaluated both the estimate $\hat{x}^{\mathrm{on}}$ computed by a weighted sum of the separated sources $\hat{s}_m$ with the predicted probabilities $\hat{y}_m$, as well as the oracle remixture of separated sources to match the on-screen and off-screen reference audios (denoted by MixIT*). In this case, notice that all methods improve the percentage of videos rated as on-screen-only from 25.7% to about 37% or 38% for all methods.

Overall, these human evaluation results suggest lower performance than the objective metrics in Table 1. One reason for this is that the binary rating task is ill-suited towards measuring variable levels of off-screen sounds. That is, a video will be rated as on-screen only if there is absolutely no off-screen sound. However, even if there is quiet off-screen sound present, or artifacts from the separation, a video will be rated as having off-screen sound. Thus, the proportion of human-rated on-screen-only videos can be interpreted as the number of cases where the model did a perfect job at removing off-screen sounds.

We plan to run new human evaluation tasks with better-matched questions. For example, we could ask raters to use a categorical scale, e.g. mean opinion score from 1 to 5. Another idea is to ask raters to score the loudness of on-screen sounds with respect to off-screen sounds on a sliding scale, where the bottom of the scale means on-screen sound is much quieter than off-screen sound, middle of the scale means on-screen sound is equal in loudness to off-screen sound, and top of the scale means on-screen sound is much louder than off-screen sound.

## A.6  PERFORMANCE ANALYSIS OF BEST MODELS

In Figure 8, we show the distributions of overall SI-SNR and SI-SNR improvement, as well as OSR for the best unsupervised and semi-supervised models. We have neglected outliers (including infinite values) in both axes in order to focus on the most common samples.

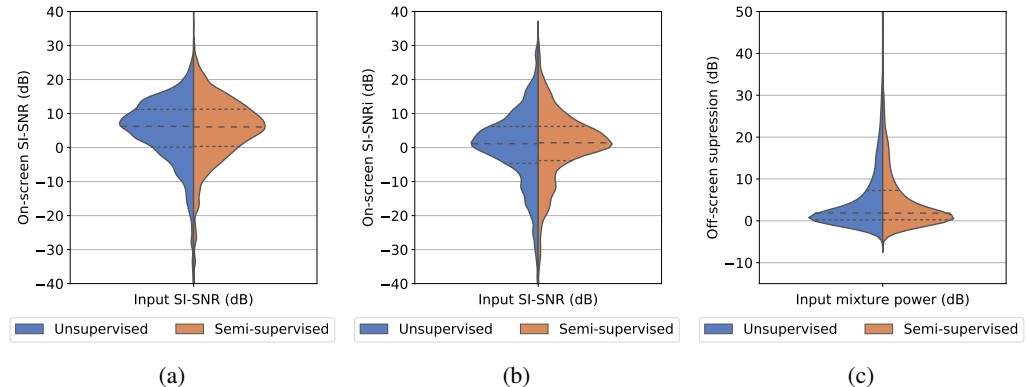

(a)                                    (b)                                    (c)

Figure 8: Distribution plots for the performance obtained by the best model in terms of on-screen SI-SNR (Figure 8a) and SI-SNRi (Figure 8b) reconstruction and off-screen power suppression (Figure 8c). The settings for the models are SOff 0% and active combinations (AC) cross-entropy loss.

In Figure 9, for on-screen MoMs we show the distribution of each performance metric for these models versus different ranges of input SI-SNRs lying between $[-30, 30]$dB, both for absolute on-screen SI-SNR (Figure 9a) and on-screen SI-SNR improvement (Figure 9b). For off-screen test MoM videos, we plot the distribution of OSR for different ranges of input mixture power lying between $[-40, 0]$dB (Figure 9c).

For on-screen SI-SNR and SI-SNRi, notice that the performance of the unsupervised and semi-supervised models is similar except for the $[-30, -20]$ dB range of input SI-SNR. In Figure 9c, note that both models achieve OSR of at least 0 dB for 75% of examples, and thus suppress off-screen sounds for at least 75% of the test data.

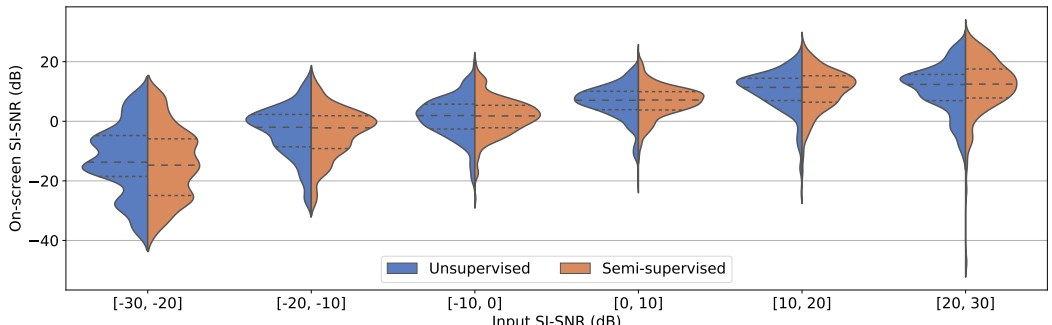

(a) On-screen reconstruction performance in terms of SI-SNR for on-screen MoMs, for each input SI-SNR bucket.

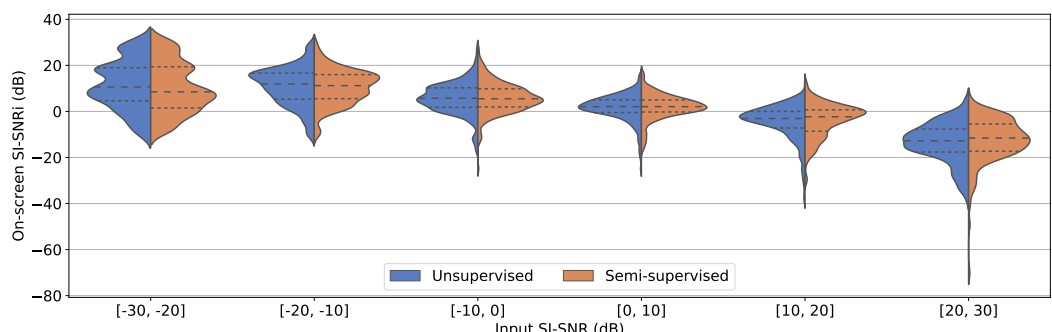

(b) On-screen reconstruction performance in terms of SI-SNR improvement (SI-SNRi) for on-screen MoMs, for each input SI-SNR bucket.

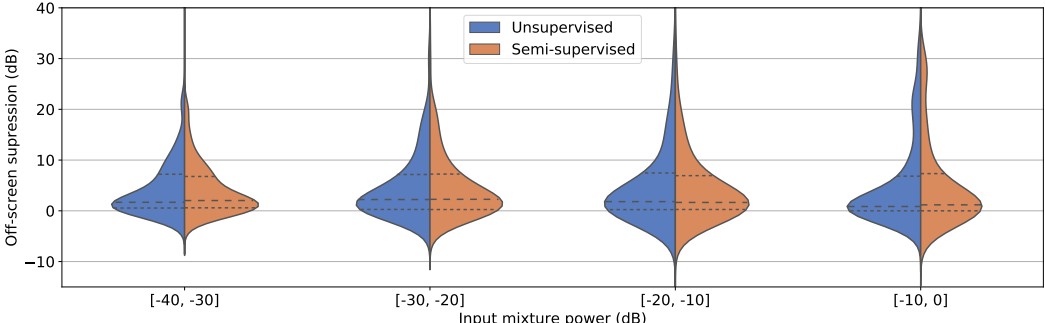

(c) Off-screen power suppression (OSR) distribution for off-screen MoMs, for each input mixture power bucket in dB.

Figure 9: Distribution plots for the performance obtained by the best model under different ranges of input SI-SNR and input mixture powers, for both unsupervised and semi-supervised settings. For each distribution plot, we depict the 25th, 50th and 75th percentiles with dashed lines. The settings for the models are SOff $0\%$ and active combinations (AC) cross-entropy loss.

## A.7 ATTRIBUTIONS

Images in figures are resized stills with or without overlaid attention maps from the following videos.

