# OpenReview forum: "Into the Wild with AudioScope: Unsupervised Audio-Visual Separation of On-Screen Sounds"
_ICLR.cc/2021/Conference — ICLR 2021 Poster_

### Official Review · AnonReviewer3 · 2020-10-26
**A new method on multi-modal source separation**

**Rating:** 6
**Confidence:** 4

**Review:**

Summary of the paper:
This paper proposes a multi-modal sound source separation framework in which they aim to separate on-screen sound. The proposed method extends the recent unsupervised source separation framework MixIT by conditioning video input. Although there have been numerous multi-modal sound source separation papers, this work goes one step further by using the sound data “in-the-wild”. The experiment results show reasonable performance on separating mixture from mixtures.



Strength:

This paper extends the existing source separation approaches to enable multi-modal source separation.



Weakness:

-----Novelty-----

The proposed method is a combination of existing approaches, which shows mere novelty.

-----Writings-----

Some unexplained notations and typos.
If the notation appears for the first time in the paper the authors must explain what it is.
For example,

1. In equation (6) please explain the notation "P(?)" in the following sentence: "minimum loss over all settings P(R) of the labels"

2. In section 4.1 what does Ms stand for?
"We use a MixIT separation loss (Wisdom et al., 2020b), which optimizes the assignment of M
estimated sources ^s = Ms (x1 + x2) to two reference mixtures x1, x2 as follows"

Some sections (especially methods) are poorly explained.
For example,
“following the same procedure as Tzinis et al. (2020)” -> following how?
“The concatenated visual embedding is resampled, fed through a dense layer, and concatenated together with all convolutional blocks.” -> resampled how? What resampling method did you use?

Furthermore, I’d like to ask the authors to specify exact neural architectures at least in Appendix.
“We also use local features extracted from an intermediate level in the visual convolutional network, that has 8 × 8 spatial locations” -> from which layer did you take the local video embedding that has the size of 8x8?

-----Experiments-----

I understand that it is hard to obtain single source on-screen clips, but it could have been better if the authors had collected some small samples and test the single mixture separation performance.


Overall:

The unsupervised separation framework is an important research direction to deal with the real-world sound sources. Although the novelty is mild, therefore, I think this work shows a promising research direction, hence recommend a weak accept.


Questions:

1.	5 video frames for 5-second of video seems too small to me. Are there any reasons for choosing such small number of frames for conditioning?

2.	Why does Global video embedding have to be an input for On-screen classifier?

---

> ### Author Response · Authors · 2020-11-16
> **Response to reviewer 3 - part 1/2**
>
> We would like to thank the reviewer for their valuable feedback on how to further improve our paper. We have extensively worked towards addressing all these comments by conducting extensive ablation studies, updating the manuscript accordingly, as well as addressing the reviewer’s concerns in this response.
>
> > The proposed method is a combination of existing approaches, which shows mere novelty.
>
> We welcome the reviewer’s concern; please see our general comment where we summarize thel findings from our ablations and concerns about the novelty of our contribution.
>
> > Typo 1: In equation (6) please explain the notation "P(?)" in the following sentence: "minimum loss over all settings P(R) of the labels"
>
> Thank you for noticing this, and pardon the typo; we accidentally commented out this notation. Added it back in text: “where $\mathcal{P}_{\geq 1} \left( \mathcal{R} \right)$ denotes the power set of indices with label of $y_i = 1$”.
>
> > In section 4.1 what does Ms stand for? "We use a MixIT separation loss (Wisdom et al., 2020b), which optimizes the assignment of M estimated sources ^s = Ms (x1 + x2) to two reference mixtures x1, x2 as follows"
>
> Please pardon the misleading notation, $\mathcal{M}_s$ was actually referring to the separation model. In our revised manuscript we have simplified the notation of the intermediate embeddings and models, now all models are denoted with $\mathcal{M}$, and the number of separated sources by $M$.
>
> >Some sections (especially methods) are poorly explained. For example, “following the same procedure as Tzinis et al. (2020)” -> following how? “The concatenated visual embedding is resampled, fed through a dense layer, and concatenated together with all convolutional blocks.” -> resampled how? What resampling method did you use?
>
> Thank you for the constructive comment, we apologize for the short description of this part of the framework. We use a simple nearest-neighbor upsampling, and added this to the text:
> “The embeddings of the video input $Z^{\mathrm{v}}_j$ can be used to condition the separation network Tzinis et al. (2020). Specifically, the image embeddings are fed through a dense layer, and a simple nearest neighbor upsampling matches the time dimension to the time dimension of the intermediate separation network activations. These upsampled and transformed image embeddings are concatenated with the intermediate TDCN++ activations and fed as input to the separation network layers.”
>
> We have also revised many sections in the paper, especially Section 3 (the main methods section), to improve our explanations, and we are happy to address any additional concerns.

---

> ### Author Response · Authors · 2020-11-16
> **Response to reviewer 3 - part 2/2**
>
>
> > I’d like to ask the authors to specify exact neural architectures at least in Appendix. “We also use local features extracted from an intermediate level in the visual convolutional network, that has 8 × 8 spatial locations” -> from which layer did you take the local video embedding that has the size of 8x8?
>
> We thank the reviewer for indicating this important detail about the reproducibility of our framework. We have added a new section, Appendix A.4, specifying the detailed architectures of the models used (separation, image embedding, audio embedding). In regards to the intermediate 8x8 convolutional map, we have extracted it from the output of the 7th separable convolutional layer block (effectively the first layer where the output dimensions for each image embedding are 8x8).  There is a specific notation with a star for the output of this layer in Table 10 in the appendix where we describe the neural network architecture of the image embedding network.
>
> > I understand that it is hard to obtain single source on-screen clips, but it could have been better if the authors had collected some small samples and test the single mixture separation performance.
>
> We agree with the reviewer that it is a hard task finding ground truth on-screen only clips with a single source. To address this, in the revision Appendix A.2 presented evaluation results with the Mandarin and AudioSet SingleSource datasets, which contain either a single speaker or single musical instrument sounds, respectively (Appendix A.2, Tables 2 and 3). Notice that despite being trained on mismatched open-domain data, our model is able to perform admirably compared to several existing approaches trained on matched data towards the specific task, some of them using full supervision. Using some oracle information to select the best sources actually improves over some of the baseline models, which indicates the efficacy of MixIT to separate these sources using primarily audio-only information.
>
> > 5 video frames for 5-second of video seems too small to me. Are there any reasons for choosing such small number of frames for conditioning?
>
> We thank the reviewer for pointing out an interesting research question that arises from the time resolution of the video frames. We chose this rate of 1 frame/second for efficiency purposes. Also, our initial premise was that we are mostly interested in the class information which might get captured from the visual stimuli, rather than synchrony. Several studies in more restrictive environments have shown that similar results could also be obtained from a single image e.g. https://arxiv.org/pdf/2007.07984.pdf. Moreover, several benchmark datasets (e.g. AudioSet-SingleSource) that we have used to evaluate the effectiveness of our approach also use a rate of 1 frame/second. Our hypothesis is that in such a diverse in-the-wild environment with a diverse set of video backgrounds, motion features would not yield significant improvement for separating audio sources compared to the class information we might obtain from a low-resolution as well. However, we are also planning on studying those interesting questions and we defer this endeavor of using higher frame rates and including motion features (e.g. optical flows Afouras et al. 2020) to future work.
>
> > Why does Global video embedding have to be an input for On-screen classifier?
>
> In our revision, we present the results of a number of ablations on the effectiveness of the global video embedding as an input to the classifier. Specifically, the contribution of the video-embedding is not critical as an input to the classifier (please see Table 4 in Appendix A.3.1).

---

### Official Review · AnonReviewer2 · 2020-10-29
**Good solution to on- vs off-screen sound classification, additional evaluation is necessary to support the full title**

**Rating:** 7
**Confidence:** 4

**Review:**

This paper describes a system for separating "on-screen" sounds from "off-screen" sounds in an audio-visual task, meaning sounds that are associated with objects that are visible in a video versus not. It is trained to do this using mixture invariant training to separate synthetic mixtures of mixtures. It is evaluated on a subset of the YFCC100m that is annotated by human raters as to whether the clips have on-screen, off-screen, or both types of sounds, with the predictions of a previously described model  (Jansen et al, 2020) helping to reduce the number with only off-screen sounds. The predictions are evaluated in terms of how well they can estimate the true on-screen sound (in terms of SI-SNR) and how well they can reject off-screen sound (in terms of a metric called off-screen suppression ratio, OSR). The results show that the system can successfully distinguish between on- and off-screen sound, but that different training regimens lead to different tradeoffs in these two metrics. The system with the best SI-SNR (8.0 dB) is trained using just data from the previous model along with the mixture invariant training criterion.

The paper presents an interesting approach to solving the on-screen vs off-screen sound problem in audio-visual source separation. While other approaches have solved similar problems for more specific source types (speech, music), this one does appear more "universal", with few assumptions tying it to a specific sound type. While this novelty is one of the strengths of the paper, it makes it more difficult to evaluate the system in comparison to established baselines. Such baselines would make it easier to understand how well the system is doing and which parts of it are the most useful and important. Perhaps it could be compared to one of the more source-specific systems on test data suited to such a system.  Another useful benchmark would be using the audio-visual coincidence prediction system of Jansen et al (2020) to assign the entire soundtrack to on or off screen and measuring the various evaluation metrics on the predictions (although infinities in the metrics make this tricky). Yet another baseline might be using an audio-only mixture of mixtures separation system, perhaps with an oracle assignment system. There is a baseline that uses oracle mixtures to compute an estimate of the relevant signals, but the opposite would also be informative as to the utility of the visual component of the system and problem.

While these additional baselines would be nice, I do not think that they are necessary for publication presently. The experiments reported are conducted thoroughly and carefully. The results, while leaving certain aspects of the optimal training program underspecified, demonstrate that the system is useful.

One additional aspect of the system that could be quantified more fully is the quality of the individual source separations. While the on- vs off-screen task is evaluated thoroughly, there is no quantitative evaluation of the source separation performance for individual sources in each mixture. It would be possible to evaluate on completely synthetic mixtures, although this is perhaps outside of the main contribution of the paper, the audio-visual combination.

Overall, this is an interesting approach that is well described. The evaluation is sufficient for the on- vs off-screen task, although not sufficient to judge whether the system has learned a completely unsupervised source separator, making the scope of the contribution somewhat more limited than it has the potential to be.

---

> ### Author Response · Authors · 2020-11-16
> **Response to reviewer 2 - part 1/2**
>
> We would like to thank the reviewer for their interest in our work and their feedback which is significant towards improving our paper. We have worked towards addressing all these comments, and please see our detailed responses below.
>
> > While this novelty is one of the strengths of the paper, it makes it more difficult to evaluate the system in comparison to established baselines. Such baselines would make it easier to understand how well the system is doing and which parts of it are the most useful and important. Perhaps it could be compared to one of the more source-specific systems on test data suited to such a system.
>
> We agree with the reviewer that in order to make a sound argument about the effectiveness of our system we should report the performance obtained from established baselines.
>
> In the revised paper, we also evaluated our AudioScope system on two existing audio-visual separation test sets within a more restricted domain, including Mandarin sentences (Hou et al 2018) and AudioSet-SingleSource (Gao and Grauman 2018, 2019), and compared our general AudioScope model to methods that are specifically tuned for these tasks. We find that non-oracle outputs of AudioScope achieve lower performance, but oracle selection or combination of sources is competitive, which indicates that the separation model is working quite well on its own to separate individual sources, and improvements for the on-screen classifier should lead to improvements for non-oracle estimates of on-screen sources.
>
> Unfortunately, it is difficult or even infeasible to run existing baselines from the literature on our particular task constructed from in-the-wild data with an open domain of sound classes. This is because these baselines from the literature often rely on supervised object detectors for specific classes and/or do not have explicit ways of dealing with off-screen sounds appearing in training videos. To attempt an approximation of these methods we ran an ablation study using a simple baseline (please see Appendix A.3.5) where the separation network performs conditional separation and has two output slots (one for the on-screen mixture and one for the off-screen one). One can view the conditional separation baseline as using a similar setup with what has been proposed by (Owens & Efros 2018), just with different architectures for the image embedding and separation networks. Moreover, we also ran the experiment with having no conditional visual information. All in all, we show that such a model totally fails to learn a good separation of on-screen and off-screen sources for our data.
>
> We also agree with the need for conducting more ablation studies in order to show which parts of our model / training scheme are the most crucial ones. Please see our general comment for a summary of findings from our ablations and concerns about the novelty of our contribution.
>
> > Another useful benchmark would be using the audio-visual coincidence prediction system of Jansen et al (2020) to assign the entire soundtrack to on or off screen and measuring the various evaluation metrics on the predictions (although infinities in the metrics make this tricky).
>
> Although the audio-visual coincidence prediction system from Jansen et al (2020) is not able to separate sounds, we agree on the importance of showing its effectiveness towards detecting whether an entire sound-track contains on-screen or off-screen sounds. We suggest that we get a rough idea of the performance of this model by checking the last paragraph of Section 5.1., where we report the on/off screen distributions before and after using the coincidence prediction system from Jansen et al (2020) on the unfiltered version of our dataset, namely:
>
> Based on human annotations, we estimate that for unfiltered data 71.3% of clips contain both on and-off-screen sounds, 2.8% contain on-screen-only sounds, and 25.9% contain off-screen-only. For the filtered data, 83.5% of clips contain on-screen and off-screen sounds, 5.6% of clips are on-screen-only, and 10.9% are off-screen-only. Thus the unsupervised filtering significantly reduced the proportion of off-screen-only clips.

---

> ### Author Response · Authors · 2020-11-16
> **Response to reviewer 2 - part 2/2**
>
> > Yet another baseline might be using an audio-only mixture of mixtures separation system, perhaps with an oracle assignment system. There is a baseline that uses oracle mixtures to compute an estimate of the relevant signals, but the opposite would also be informative as to the utility of the visual component of the system and problem.
>
> We thank the reviewer for bringing this point forward which we also think is an important ablation for audio-visual separation systems. In Table 1, column “MixIT” refers to the oracle assignment of the audio-based MixIT separation, where the only use of visual information is the conditioning of the separation network on image embeddings. We ran an ablation of this visual information conditioning and reported the results in Table 4 in Appendix A.3.1. We observed a moderate drop in audio separation performance. Moreover, we also ran an ablation study with a purely audio-based system that predicts the on and off screen sources in Table 8 in Appendix A.3.5, where we see that even with optimal assignment, the audio-only system is not able to separate the off-screen from on-screen sounds.
>
> > One additional aspect of the system that could be quantified more fully is the quality of the individual source separations. While the on- vs off-screen task is evaluated thoroughly, there is no quantitative evaluation of the source separation performance for individual sources in each mixture. It would be possible to evaluate on completely synthetic mixtures, although this is perhaps outside of the main contribution of the paper, the audio-visual combination.
>
> This is a very interesting aspect of our model that we wish we had more space in the paper in order to elaborate on. We would like to mention that our whole system is based on top of MixIT (Wisdom et al. 2020b) which has shown strong performance on per-source universal sound source separation with a purely audio based approach. Considering that AudioScope employs these separated sources and also that we stop the gradient from the classifier loss back to MixIT, we would like to suggest that we have an adequate universal sound separation system even for single sources. In addition, our new version includes results where we have also run our model on the Mandarin sentences and AudioSet-ingleSource datasets, which contain human-annotated single-source references.
>
> > Overall, this is an interesting approach that is well described. The evaluation is sufficient for the on- vs off-screen task, although not sufficient to judge whether the system has learned a completely unsupervised source separator, making the scope of the contribution somewhat more limited than it has the potential to be.
>
> We thank the reviewer for their interest in our work and providing useful feedback for improving the paper. We hope that our comment responses above and our newest ablation studies assuage concerns about the effectiveness of AudioScope for separating single sources from mixtures. We are happy to further address any additional comments.

---

### Official Review · AnonReviewer1 · 2020-10-29
**Official Blind Review #1**

**Rating:** 6
**Confidence:** 4

**Review:**

This paper proposed an unsupervised method for open-domain, audio-visual separation system. The proposed model was optimized using the newly suggested mixture invariant training (MixIT) together with a cross entropy loss function. The authors suggest to separately process the audio and video, and next align them with a spatiotemporal attention module.

Unsupervised source separation, especially for open-domain is an interesting and important research direction. However, there are several concerns with this submission that need to be addressed first.

My main concern is the contribution of this paper. The authors presented a fairly complicated system comprised of several modules. I would expect the authors to run an ablation study / analysis to better understand their contribution to the final model performance. For instance, why do we need attentional pooling? do we need it in both audio and video? When does the model fail? can we learn something from it?

Second, I know the authors said this is the first system to do so, however, I would still expect the authors to compare to some baseline. Maybe a fully supervised one? Otherwise it is hard to interpret the numbers presented in Table 1. It is hard to understand how good is this system and how much room for improvement do we have.

Regarding the samples, it is a bit hard to interpret this results. For every file there are 5 videos and 5 separated samples, some of them sound almost identical. Again, there is no baseline to compare against, so it is hard to understand how good the quality of the separations is. I suggest the authors to improve the samples page to better present emphasis their results.

A question for the authors, since you treated this task at an unsupervised task, did you try to run some subjective evaluations? maybe let users annotate the sound files and compare variance?

---

> ### Author Response · Authors · 2020-11-16
> **Response to reviewer 1**
>
> We would like to thank the reviewer for their constructive feedback, we feel that working to address these concerns have greatly improved the paper. We think we have addressed all or most of the concerns in the updated version of our paper, as well as the newly updated supplementary material. Please see our specific responses below.
>
> > The authors suggest to separately process the audio and video, and next align them with a spatiotemporal attention module.
>
> This is essentially true, though we would like to mention that the audio source separation network is also conditioned on video. Thus, the audio-visual framework does have interdependencies in terms of how audio and visual information are being processed. We added some discussion of our architecture design choices, as well as ablations. Also, we have revised Figure 2 and its description to be more clear, which we hope makes our system architecture easier to understand.
>
> >My main concern is the contribution of this paper. The authors presented a fairly complicated system comprised of several modules. I would expect the authors to run an ablation study / analysis to better understand their contribution to the final model performance. For instance, why do we need attentional pooling? do we need it in both audio and video? When does the model fail? can we learn something from it?
>
> We welcome this comment, and we hope that the new ablation studies we have performed helped answer these questions. Please see our general comment section where we summarize our findings from our ablation study and address the concerns about the novelty of our contribution.
>
> Regarding the question about the attentional pooling mechanism, one of our ablations replaced the attentional pooling operation with a simple mean-pooling operation across time. The results are in Appendix A.3.2, and show that using mean pooling achieves comparable classification and on-screen SI-SNR, but reduces off-screen suppression.
>
> >I know the authors said this is the first system to do so, however, I would still expect the authors to compare to some baseline. Maybe a fully supervised one? Otherwise it is hard to interpret the numbers presented in Table 1. It is hard to understand how good is this system and how much room for improvement do we have.
>
> We agree with the reviewer that in order to make a sound argument about the effectiveness of our system we should report the performance obtained from simpler baselines. To address this, we ran an ablation study using a simple baseline separation model (please see Appendix A.3.5) that performs conditional separation with two output slots (one for the on-screen mixture and one for the off-screen one). One can view this baseline as effectively using a similar setup with what has been proposed by Owens & Efros 2018. Note that the proposed AudioScope model substantially outperforms this baseline.
>
> Furthermore, in Appendix A.2 we evaluate AudioScope on two audio-visual separation test sets from the literature and compare them to state-of-the-art baselines from the literature: Mandarin sentences for audio-visual speech enhancement, and AudioSet-SingleSource for audio-visual musical instrument separation. Non-oracle settings of AudioScope that use predicted classifier probabilities do not outperform the baselines, but oracle settings (i.e. selecting best source, or finding best mixing of sources to references) beat state-of-the-art. This is somewhat remarkable, given AudioScope is trained on open-domain YFCC100m data, and the baselines from the literature are trained specifically for these tasks. This also suggests that improving the performance of the classifier could lead to non-oracle solutions that are competitive with state of the art.
>
> >Regarding the samples, it is a bit hard to interpret this results. For every file there are 5 videos and 5 separated samples, some of them sound almost identical. Again, there is no baseline to compare against, so it is hard to understand how good the quality of the separations is. I suggest the authors to improve the samples page to better present emphasis their results
>
> To address this, we have updated our supplementary material to make them clearer. We hope that the new version is easier to understand and better conveys the effectiveness of our method.
>
> >A question for the authors, since you treated this task at an unsupervised task, did you try to run some subjective evaluations? maybe let users annotate the sound files and compare variance?
>
> We thank the reviewer for underlining the importance of subjective human evaluations for a method like the one we propose. Indeed, we are currently running a large-scale human evaluation, where human annotators rate the output video for presence of on-screen sounds and off-screen sounds. We should be able to share the results of this study soon.

---

> > ### Comment · AnonReviewer1 · 2020-11-24
> > **Updated reivew**
> >
> > I would like to thank the authors for providing additional experiments and results. I appreciate the authors hard work.
> > After reading the other reviews, the authors response and the updated manuscript, I find the additional results and ablation study interesting. Indeed, the ablation provided by the authors shed more light on some model components, however I still feel that the proposed model is sort a composition of existing building blocks w. no much intuition to justify it.
> >
> > I am willing to increase my score. The final score will be based on the evaluation of all reviewers' comments and the authors' response.

---

> > > ### Author Response · Authors · 2020-11-24
> > > **Response to updated review from R1**
> > >
> > > We would like to thank the reviewer for acknowledging our effort towards refining the manuscript. We apologize if the selection of the model’s components still seems somewhat arbitrary; in our manuscript revisions we tried to add some explanation of our decisions about the overall architecture, as well as in our first general comment to all the reviewers. We would like to mention that the focus of the paper is not on the architecture of the building blocks -- the image embedding block, the sound separation block and the audio embedding block.  Indeed these are chosen to represent reasonable choices based on their successful prior use in the literature.  The specific architecture within these blocks is not a novel contribution, and it's entirely possible that they can be improved upon.  Rather, our main novel contribution is the overall system architecture, as well as the open-domain and unsupervised training method, built on top of mixture invariant training, which, to the best of our knowledge, is the first audio-visual separation system that can be trained on open-domain, in-the-wild video data, and this training scheme could be used with any choice of architecture for the component blocks.
> > >
> > > We would welcome any additional comments, and thank you again for your useful and detailed comments that helped improve the manuscript.

---

> ### Author Response · Authors · 2020-11-23
> **Response to reviewer 1 after second revision**
>
> >>R4 comment: A question for the authors, since you treated this task at an unsupervised task, did you try to run some subjective evaluations? maybe let users annotate the sound files and compare variance?
>
> >Our initial response: We thank the reviewer for underlining the importance of subjective human evaluations for a method like the one we propose. Indeed, we are currently running a large-scale human evaluation, where human annotators rate the output video for presence of on-screen sounds and off-screen sounds. We should be able to share the results of this study soon.
>
> As initially promised in our response last week, we added the results from a human evaluation study in the new Appendix A.5 in the second revision.

---

### Official Review · AnonReviewer4 · 2020-10-29
**Some thoughts**

**Rating:** 6
**Confidence:** 4

**Review:**

**Pros**

Audio-visual sound source separation is an impotant task. The paper pushes the boundary from specific domains (e.g. speakers, musics, etc) to generalized open-domain, which is crucial and far from trivial.

The authors introduced a new, large-scale, open-domain dataset for on-screen audio-visual separation. The videos span 2500 hours, 55 of which are verified by human labelers. The dataset will definitely be very useful for the community as it is way more diverse than before.

---
**Cons**

*Related work*

To my understanding, Owens and Efros (2018) did not assume fixed number of speakers. While they validate their method under such setting, there is actually no limitation in their model that prevents them to have multiple on-screen sources. Therefore, I'm not sure about the first contribution, except the "open-domain" part.

*Model*

In terms model architecture, (maybe I have missed something but) I didn't see much novelty in the current state. To me, the proposed model is simply a composition of multiple exisiting modules from previous work. Please note that I'm not saying building upon the sucess of previous efforts are wrong by any means. I just had the feeling that the authors are piecing various building blocks together w/o providing much intuition. Maybe there is some novelty lying within the current design. For instance, the authors may have developed a novel routing/module drawing inspiration from a certain observation; the combination of xxx and yyy is based on deliberate choice. It is, however, not clear to me at this point, at least the writing does not reflect it.

Furthermore, if the network is the key player in this paper, the authors shall provide more evidence. While the authors do show conduct some ablation studies on the losses and data, there aren't any analysis regarding the importance of each module (e.g. how critical is the attention design?). It is thus difficult for readers to understand which part of the network is crucial for the success, and what is the major novelty within the architecture. The current form provides very litte intuition and take away.

*Objectives*

Eq. 4 and Eq. 6 looks very similar to me. Aren't they equivalent if the **A** in Eq. 6 is the same as **A** that minimizes Eq. 2, since the assignment in Eq. 4 comes from Eq. 2? On the other side, if the two **A** are different, what's the intuition of exploiting different A for different loss?


*Writing/Presentation*

The flow of the model architecture section can be improved. The authors did not provide any high-level context. Instead, they simply dig into the " details" of each module. I'm not aware of the connections among while reading the text. Instead, I need to constantly check the figure and infer these.

I also don't know what is the input/output of the model and what representations they are using. Shoudn't these be explained at the very beginning? These are not explicitly defined and I need to infer them myself. For instance, is the output of masking network a M x T soft mask with values lying within [0, 1]? do they exploit waveform (fig. 2) or spectrogram (fig. 1) for audio? I figured/inferred a lot these out after I moved to the experiment section. But from two cents, these are related to the model.

*Experiments*

Currently the authors only evaluate the model on their own dataset. How does the model work on existing datasets? For instance, AudioSet, MUSIC, FAIR-Play, etc. It seems that there is nothing preventing them from applying their model to those datasets. Without these results, it would be hard to justify if the proposed "open-domain model" can generalize to "a specific domain." I think at least  direct inference (generalization) or train from scratch is required.

Furthremore, the authors did not compare with any baseline. It seems to me that quite a few prior art [Owens and Efros (2018), Hang et al (2018), etc] can serve as baselines with minor modification. Take Owens and Efros (2018) for example. While they may not be able to decompose each sound source within the on-screen mixture, one can still leverage it to evaluate the on/off-screen separation. The authors thus shall be able to report SI-SNR too. Otherwise its very difficult for people to do an apple-to-apple comparison of this work and prior work.

The authors should report more performance at more percentiles. The most illustrative way is to show the cumulative plot - how many % of data have error less than x.

Is there an intuition of why only pre-training part of the model? Why not pre-train the separation network too?

---
**Minor comments**

How do the authors define the diversity (Sec. 5.1) of the videos? Do they make use of the tags provided by YFCC100M? Also, whats the statistics of those filtered data? Would be great to provide more details so that we know its indeed covering a wide range of semantic categories.

Some relevant literatures are missing. For instance, [1] also associates the visual information with speeching signal. The subjectives (eg person, dog, birds) in the paper can be viewed as on-screen audio, while prepositions can be seen as off-screen voice. There are definitely a lot more on this direction, but this paper pop out my head right away.

[1] Jointly Discovering Visual Objects and Spoken Words from Raw Sensory Input. ECCV 2018.

---

> ### Author Response · Authors · 2020-11-16
> **Response to reviewer 4 - part 1/3**
>
> We thank the reviewer for their constructive feedback for improving the clarity of our presentation and the soundness of our work. We hope that our revisions and specific responses below help address the reviewer’s concerns.
>
> > Furthremore, the authors did not compare with any baseline. It seems to me that quite a few prior art [Owens and Efros (2018), Hang et al (2018), etc] can serve as baselines with minor modification. Take Owens and Efros (2018) for example. While they may not be able to decompose each sound source within the on-screen mixture, one can still leverage it to evaluate the on/off-screen separation. The authors thus shall be able to report SI-SNR too. Otherwise its very difficult for people to do an apple-to-apple comparison of this work and prior work.
> > To my understanding, Owens and Efros (2018) did not assume fixed number of speakers. While they validate their method under such setting, there is actually no limitation in their model that prevents them to have multiple on-screen sources. Therefore, I'm not sure about the first contribution, except the "open-domain" part.
>
> Thanks for these comments. In order to address the comparison with a model resembling (Owens & Efros 2018), we ran an ablation study using a model which resembles the aforementioned system (please see Appendix A.3.5 in the revised paper). The separation network performs conditional separation with two output slots (one for the on-screen mixture and one for the off-screen one), exactly the same as the model proposed by Owens & Efros 2018, except with different network architectures for the image embedding and  separation networks. Overall, we show that such a model totally fails to learn a good separation of on-screen and off-screen sources for our data.
>
> Regarding variable number of in-the-wild sources with the framework proposed by (Owens & Efros 2018), it’s a good point that their architecture is not constrained to fixed number of sources, but since the training and testing was performed using a fixed number of two speakers, we don’t know how well it works for variable sources. Moreover, the model is trained assuming all the speakers appear on-screen, which is certainly not the case for in-the-wild data, where a significant percentage of training videos contain sources that are not on-screen. We revised the paper to reflect this. In the revised version, we showed that a baseline model, which is similar to (Owens & Efros 2018), exhibits poor performance for reconstructing on-screen sources and suppressing the off-screen ones.  Regarding the novelty of our contribution, please see the discussion of novelty in our general comment.
>
> > In terms model architecture, (maybe I have missed something but) I didn't see much novelty in the current state. To me, the proposed model is simply a composition of multiple exisiting modules from previous work. Please note that I'm not saying building upon the sucess of previous efforts are wrong by any means. I just had the feeling that the authors are piecing various building blocks together w/o providing much intuition. Maybe there is some novelty lying within the current design. For instance, the authors may have developed a novel routing/module drawing inspiration from a certain observation; the combination of xxx and yyy is based on deliberate choice. It is, however, not clear to me at this point, at least the writing does not reflect it. Furthermore, if the network is the key player in this paper, the authors shall provide more evidence. While the authors do show conduct some ablation studies on the losses and data, there aren't any analysis regarding the importance of each module (e.g. how critical is the attention design?). It is thus difficult for readers to understand which part of the network is crucial for the success, and what is the major novelty within the architecture. The current form provides very litte intuition and take away.
>
> We welcome the reviewer’s concern; please see our general comment where we address all findings from our ablations and concerns about the novelty of our contribution. Moreover, we have refined the description of our model architecture selection which would hopefully alleviate some of your concerns about why we made certain design choices. In particular, we emphasize that we don’t think the main novelty of our paper lies in the architectural choices, but rather in building on top of MixIT to train an open-domain audio-visual separation system on in-the-wild data. Please see the updated Section 3, and we are happy to further revise it if more description or discussion is needed.

---

> ### Author Response · Authors · 2020-11-16
> **Response to reviewer 4 - part 2/3**
>
> > Eq. 4 and Eq. 6 looks very similar to me. Aren't they equivalent if the A in Eq. 6 is the same as A that minimizes Eq. 2, since the assignment in Eq. 4 comes from Eq. 2? On the other side, if the two A are different, what's the intuition of exploiting different A for different loss?
>
> We apologize for any misconception caused by our notation. One possible reason for confusion was the usage of $\mathcal{A}$ (replaced with $\mathcal{S}$ in the revision for clarity) in eq (6) is an element of the powerset of positive indices $\mathcal{R}$ from the optimal mixing matrix $A$ found in eq 2. Note that (2) is minimizing a signal-level SNR objective over all possible mixing matrices $A$, and (6) is minimizing cross-entropy over all subsets of the positive labels in the optimal mixing matrix $A^*$ found in (2). We have our revised our description of these losses to make it more clear.
>
> > The flow of the model architecture section can be improved. The authors did not provide any high-level context. Instead, they simply dig into the " details" of each module. I'm not aware of the connections among while reading the text. Instead, I need to constantly check the figure and infer these.
> >  I also don't know what is the input/output of the model and what representations they are using. Shoudn't these be explained at the very beginning? These are not explicitly defined and I need to infer them myself. For instance, is the output of masking network a M x T soft mask with values lying within [0, 1]? do they exploit waveform (fig. 2) or spectrogram (fig. 1) for audio? I figured/inferred a lot these out after I moved to the experiment section. But from two cents, these are related to the model.
>
> Thank you for these comments on improving the architecture section, and we apologize for any misunderstanding caused by lack of clarity in our presentation. We have refined the description of our model architecture and updated Figure 2 with labels for intermediate embeddings which serve as inputs/outputs of the consisting modules.
>
> As depicted in Figure 2, the output of the separation network are the raw waveforms of the estimated sources. To predict these waveforms, the separation network internally produces a mask with elements between [0, 1] for each source. These masks are produced in the learnable analysis coefficients domain (similar to a spectrogram, although the decomposition bases are learnable) and are multiplied element-wise with the analysis coefficients of the input waveform. Waveforms are reconstructed from the masked coefficients using a transpose Conv1D layer. For more information about these internal network activations please see our new Appendix A.4 on network architectures. Finally, we do not use spectrogram representations anywhere in our model; these are purely for visualization purposes.
>
> We hope these changes make the model easier to understand and alleviate some of the reviewer’s concerns about clarity of the description. We are happy to further refine this description, as requested.
>
> > Currently the authors only evaluate the model on their own dataset. How does the model work on existing datasets? For instance, AudioSet, MUSIC, FAIR-Play, etc. It seems that there is nothing preventing them from applying their model to those datasets. Without these results, it would be hard to justify if the proposed "open-domain model" can generalize to "a specific domain." I think at least direct inference (generalization) or train from scratch is required.
>
> We welcome the reviewer’s concern about evaluating our model on existing benchmark datasets; please see our updated Appendix A.2 containing the results from direct inference of AudioScope on AudioSet-SingleSource and Mandarin speech enhancement datasets.
>
> We would like to mention that the FAIRPlay https://arxiv.org/pdf/1812.04204.pdf dataset has been introduced in order to synthesize binaural recording from single mixture recordings. In order to evaluate our model with FAIRPlay we have to add synthetic noise, which lies outside of the scope of this paper. The test set used for the MUSIC dataset https://arxiv.org/pdf/1804.03160.pdf is created by choosing 10 random videos from the initial dataset, but the exact video identifiers are not defined. We are actively working towards contacting the authors so we can fairly perform this evaluation.

---

> ### Author Response · Authors · 2020-11-16
> **Response to reviewer 4 - part 3/3**
>
> > The authors should report more performance at more percentiles. The most illustrative way is to show the cumulative plot - how many % of data have error less than x.
>
> We welcome the reviewer’s comment towards refining the visualization of our results. Although the cumulative distribution plot can be easily obtained from the two-dimensional scatter plots in Figure 3 using a projection towards the y-axis, we suggest that the current plots depict a more holistic view of the system’s performance, considering that input SNR is highly correlated with SI-SNR performance. For instance, a source with extremely low input SNR, nearly impossible to hear, would be challenging for the model and perhaps result in poor performance, but a high input SNR source should be much easier to separate and thus result in good performance. However, in a one-dimensional cumulative distribution plot, both points would collapse to the same percentile. We are exploring better ways of reporting performance across more percentiles.
>
> > Is there an intuition of why only pre-training part of the model? Why not pre-train the separation network too?
>
> We thank the reviewer for bringing up an interesting research question that arises from the usage of pre-trained modules for the image embedding, audio embedding and the separation network. From the ablations we ran, we observed a significant boost in terms of separation and detection performance by using pre-trained embedding networks compared to training them from scratch (please see Table 5 in the Appendix). Our intuition for not pretraining the separation model is as follows. For the embedding networks, we do not include the contrastive loss more oriented towards audio-visual coincidence regardless of on-screen/off-screen presence, instead of explicitly identifying on-screen sounds. In contrast, we do use the MixiT loss during training of AudioScope, and since the MixIT loss is at the audio sample level and thus has a label for each audio sample, we suspect it’s quite a lot stronger than the classification loss, which only has a label per utterance. Thus, we don’t expect that pretraining the separation network with MixIT will have much effect, but to verify this intuition, we will plan to run an ablation where we pretrain the separation network with MixIT.
>
> > How do the authors define the diversity (Sec. 5.1) of the videos? Do they make use of the tags provided by YFCC100M? Also, whats the statistics of those filtered data? Would be great to provide more details so that we know its indeed covering a wide range of semantic categories.
>
> We agree with the reviewer that we need to further investigate the metadata of the widely used YFCC100M dataset http://projects.dfki.uni-kl.de/yfcc100m/globalstats that would provide a better quantitative measure of the diversity. However, the videos contain user tags from free written speech which make the extraction of those statistics not trivial. Studies which are explicitly focused on working with those user tags also claim that the initial set of the YFCC100M videos contain a diverse set of classes and also provide some visualizations [1]. We are actively working towards trying to extract some statistics and trying to identify a representative ontology of the sound classes.
>
> [1] Xian, Y., Korbar, B., Douze, M., Schiele, B., Akata, Z. and Torresani, L., 2020. Generalized Many-Way Few-Shot Video Classification. arXiv preprint arXiv:2007.04755. https://arxiv.org/pdf/2007.04755.pdf
>
> > Some relevant literatures are missing. For instance, [1] also associates the visual information with speeching signal. The subjectives (eg person, dog, birds) in the paper can be viewed as on-screen audio, while prepositions can be seen as off-screen voice. There are definitely a lot more on this direction, but this paper pop out my head right away. [1] Jointly Discovering Visual Objects and Spoken Words from Raw Sensory Input. ECCV 2018.
>
> Thanks for bringing this work to our attention. Though it does not directly concern audio-visual on-screen separation,  it is relevant to the spatiotemporal attention we use, and we have added a reference in the model architecture description (Section 3). Namely, “These audio embeddings are then pooled over time and used in the audio-visual spatio-temporal attention network to retrieve, for each source, a representation of the visual activity that best matches the audio, similar to the associative maps extracted from the internal network representations proposed by Harwath et al. (2018).” We are happy to include any missing citations that we might have accidentally omitted.

---

> ### Author Response · Authors · 2020-11-23
> **Response to reviewer 4 after second revision**
>
> >>R4 comment: Currently the authors only evaluate the model on their own dataset. How does the model work on existing datasets? For instance, AudioSet, MUSIC, FAIR-Play, etc. It seems that there is nothing preventing them from applying their model to those datasets. Without these results, it would be hard to justify if the proposed "open-domain model" can generalize to "a specific domain." I think at least direct inference (generalization) or train from scratch is required.
>
> >Our initial response: We welcome the reviewer’s concern about evaluating our model on existing benchmark datasets; please see our updated Appendix A.2 containing the results from direct inference of AudioScope on AudioSet-SingleSource and Mandarin speech enhancement datasets.
> We would like to mention that the FAIRPlay https://arxiv.org/pdf/1812.04204.pdf dataset has been introduced in order to synthesize binaural recording from single mixture recordings. In order to evaluate our model with FAIRPlay we have to add synthetic noise, which lies outside of the scope of this paper. The test set used for the MUSIC dataset https://arxiv.org/pdf/1804.03160.pdf is created by choosing 10 random videos from the initial dataset, but the exact video identifiers are not defined. We are actively working towards contacting the authors so we can fairly perform this evaluation.
>
> In the second revision, we added an evaluation of AudioScope on the MUSIC dataset, as requested by the reviewer. Please see our updated Appendix A.3.2. Table 4.
>
> >>R4 comment: The authors should report more performance at more percentiles. The most illustrative way is to show the cumulative plot - how many % of data have error less than x.
>
> >Our initial response: We welcome the reviewer’s comment towards refining the visualization of our results. Although the cumulative distribution plot can be easily obtained from the two-dimensional scatter plots in Figure 3 using a projection towards the y-axis, we suggest that the current plots depict a more holistic view of the system’s performance, considering that input SNR is highly correlated with SI-SNR performance. For instance, a source with extremely low input SNR, nearly impossible to hear, would be challenging for the model and perhaps result in poor performance, but a high input SNR source should be much easier to separate and thus result in good performance. However, in a one-dimensional cumulative distribution plot, both points would collapse to the same percentile. We are exploring better ways of reporting performance across more percentiles.
>
> In the second revision, we added Appendix A.6, which shows a more detailed performance analysis using more percentiles for the best unsupervised and the best semi-supervised models. Please see our updated Appendix A.6., Figures 8 and 9.

---

### Author Response · Authors · 2020-11-16
**General comment to all reviewers**

We would like to thank all the reviewers for their interest in our work and for providing constructive feedback. We have worked towards addressing all their comments by conducting extensive ablation studies, updating the manuscript according to their comments, revising the supplementary material, and addressing specific concerns in each individual follow-up response. We hope that these changes serve as improvements to the clarity of our presentation and the completeness of our work.
### Ablations for components of the system
All of the reviewers expressed concerns about our fairly complex system and how each component affects system performance. In our updated version of the paper we have augmented the appendix with a large set of ablation studies in order to shed light on specific architectural choices and the contribution of individual components of the system (please see Appendix A.3 of the revised paper). We present here a small summary of our findings, which are also described in the revised appendix. The following aspects seem to alter the performance of our system in the following way:
* Excluding the visual conditioning of the audio separation network results in a decrease of about 2 dB in terms of on-screen SI-SNR, which suggests that it is useful, but not essential (see Appendix A.3.1, Table 4).
* Not using the global video embedding or the audio embeddings at the classifier input seems more or less insignificant for unsupervised training, except for a drop in off-screen separation. For semi-supervised training, removing these inputs decreases AUC-ROC and on-screen SI-SNR by a fair amount (see Appendix A.3.1, Table 4).
* Training the embedding networks from scratch seems to have a severe negative impact on performance under all metrics (see Appendix A.3.1, Table 4).
* Despite our initial hypothesis that the network would not be able to learn a good solution if data are not filtered by the unsupervised coincidence model, our ablation study suggests that for unsupervised training, our proposed active combinations loss function is robust to this label noise, and thus a larger amount of data with noisier labels is better than a smaller amount of data with less noisy labels (see Appendix A.3.3, Table 6).
* Increasing the number of output sources to 6 or 8 seems to only shift the operating point of our system by pushing it towards suppressing more power from the off-screen sources and lowering the fidelity of reconstruction fidelity for the on-screen ones. We hypothetize that this is because the separation model is getting more prone to over-separation and the classifier is becoming biased towards predicting lower audio-visual correspondence scores (see Appendix A.3.4, Table 7).
* Replacing the attentional pooling operation with a simple average operation achieves comparable classification and on-screen SI-SNR, but reduces off-screen suppression (see Appendix A.3.2, Table 3).

We have also conducted a large ablation study using baseline models (see Appendix A.3.5) and different datasets in order to be more easily comparable to the literature (see Appendix A.2.).

### Novelty
All reviewers expressed concerns about the novelty of our work. We would like to suggest that the main contribution of our paper does not rely on the specific selections of the independent modules of our proposed audio-visual separation and classification framework. Rather, we think the main novelty lies in the following aspects of our self-supervised training setup:

* We base the success of our approach on the effectiveness of the MixIT audio separation network, rather than depending on object detection systems that previous audio-visual separation methods generally rely on.
* We show that MixIT assignments can be used as pseudo-labels to train an audio-visual classifier to identify which sources appear on-screen.
* We introduce the usage of multiple instance and active combinations losses in order to compensate for the noisy labels that we get from MixIT assignment.
* We propose new ways of constructing mixtures of mixtures in order to train our system with in-the-wild data.

Our proposed framework might not be the only possible solution to the problem, but to the best of our knowledge it is the first method that provides a solution to unsupervised audio-visual information for all types of sounds.  Our evaluations on more specific, restricted domains, i.e. speech enhancement and music source separation (please see Appendix A.2) also show that our system can effectively perform on-screen sound separation without any finetuning. We anticipate that fine-tuning on these more specific domains would further improve performance.

---

### Author Response · Authors · 2020-11-23
**General comment to all reviewers after second revision**

We have uploaded a second revision of our paper with the following additional changes. We think that these additional changes address outstanding reviewer comments. Overall, we are very grateful to the reviewers for their thoughtful and useful comments, as we think they have greatly improved the paper. We believe that our changes have addressed all outstanding reviewer concerns, but we also welcome any additional comments.

Changes in second revision:
* We evaluated our models on the MUSIC dataset, and compared them to baselines from the literature. Overall we observe similar results to our evaluation on AudioSet-SingleSource. Please see new Appendix A.3.2.
* We conducted a human evaluation task. In retrospect, we found that this human evaluation was a bit strict for our task, in that raters were very sensitive to any presence of off-screen sound. We include some discussion of improved ideas we would like to try for our next human evaluations that we plan to run. Please see new Appendix A.5.
* We included a more detailed analysis of performance by making violin plots versus bins of input SI-SNR or power, which include percentiles and visualizations of the distribution. Please see new Appendix A.6.

---

### Decision · Program_Chairs · 2021-01-07
**Final Decision**

**Decision:**

Accept (Poster)

**Comment:**

This paper presents a new, large-scale, open-domain dataset for on-screen audio-visual separation, and provides an initial solution to this task. As the setting is quite specialized, the authors proposed a neural architecture based on spatial-temporal attentions (while using existing learning objective for audio separation). The reviewers were initially concerned that, while reasonably motivated, the architecture seemed some arbitrary. The authors then provided extensive ablation studies to evaluate the significance of each component with existing datasets, and these efforts are appreciated by reviewers. The authors may consider re-organizing the paper and moving some ablation studies to the main text. On the other hand, the reviewers believe that the dataset will be very useful for the community due to its diversity in content and label quality.